# Hybrid optimal-FOPID based UPQC for reducing harmonics and compensate load power in renewable energy sources grid connected system

T. Anuradha Devi[1], G. Srinivasa Rao[2], T. Anil Kumar[3], B. Srikanth Goud[3], Ch. Rami Reddy[4,5]*, Mbadjoun Wapet Daniel Eutyche[6], Flah Aymen[5,7,8,9,10,11], Claude Ziad El-Bayedh[12], Habib Kraiem[13], Vojtech Blazek[11]

1 Electrical and Electronics Engineering, Vardhaman College of Engineering, Hyderabad, India, 2 Electrical and Electronics Engineering, CMR College of Engineering & Technology, Hyderabad, India, 3 Electrical and Electronics Engineering, Anurag University, Hyderabad, India, 4 Electrical and Electronics Engineering, Joginpally B R Engineering College, Hyderabad, India, 5 Applied Science Research Center, Applied Science Private University, Amman, Jordan, 6 Department of Electrical and Telecommunications Engineering, National Advanced School of Engineering of Yaoundé, University of Yaounde, Cameroon, 7 Energy Processes Environment and Electrical Systems Unit, National Engineering School of Gabès, University of Gabes, Gabes, Tunisia, 8 University of Business and Technology (UBT), College of Engineering, Jeddah, Saudi Arabia, 9 MEU Research Unit, Middle East University, Amman, Jordan, 10 University of Gabes, Private Higher School of Applied Sciences and Technology of Gabes, Gabes, Tunisia, 11 ENET Centre, VSB —Technical University of Ostrava, Ostrava, Czech Republic, 12 Department of Electrical Engineering, Bayeh Institute, Amchit, Lebanon, 13 Department of Electrical Engineering, College of Engineering, Northern Border University, Arar, Saudi Arabia

* crreddy229@gmail.com

**Data Availability Statement:** All data are in the paper.

## Abstract

Integration of renewable energy sources (RES) to the grid in today's electrical system is being encouraged to meet the increase in demand of electrical power and also overcome the environmental related problems by reducing the usage of fossil fuels. Power Quality (PQ) is a critical problem that could have an effect on utilities and consumers. PQ issues in the modern electric power system were turned on by a linkage of RES, smart grid technologies and widespread usage of power electronics equipment. Unified Power Quality Conditioner (UPQC) is widely employed for solving issues with the distribution grid caused by anomalous voltage, current, or frequency. To enhance UPQC performance, Fractional Order Proportional Integral Derivative (FOPID) is developed; nevertheless, a number of tuning parameters restricts its performance. The best solution for the FOPID controller problem is found by using a Coati Optimization Algorithm (COA) and Osprey Optimization Algorithm (OOA) are combined to make a hybrid optimization CO-OA algorithm approach to mitigate these problems. This paper proposes an improved FOPID controller to reduce PQ problems while taking load power into account. In the suggested model, a RES is connected to the grid system to supply the necessary load demand during the PQ problems period. Through the use of an enhanced FOPID controller, both current and voltage PQ concerns are separately modified. The pulse signal of UPQC was done using the optimal controller, which analyzes the error value of reference value and actual value to generate pulses. The integrated

**Funding:** The authors extend their appreciation to the Deanship of Scientific Research at Northern Border University, Arar, KSA, for funding this research work through project number NBU-FFR-2024-2484-02, also the authors declare that the article has been produced with the financial support of the European Union under the REFRESH – Research Excellence For Region Sustainability and High-tech Industries project number CZ.10.03.01/00/22_003/0000048 via the Operational Programme Just Transition and paper was supported by the following project TN02000025 National Centre for Energy II awarded to HK.

design mitigates PQ issues in a system at non-linear load and linear load conditions. The proposed model provides THD of 12.15% and 0.82% at the sag period, 10.18% and 0.48% at the swell period, and 10.07% and 1.01% at the interruption period of non-linear load condition. A comparison between the FOPID controller and the traditional PI controller was additionally taken. The results showed that the recommended improved FOPID controller for UPQC has been successful in reducing the PQ challenges in the grid-connected RESs system.

## 1. Introduction

Energy demand has been rising recently, which has resulted in an increase in greenhouse gas emissions [1]. RES are regarded as the most effective way to lower greenhouse gas emissions while increasing energy production. Power can be produced from renewable energy sources without polluting the environment with hazardous emissions [2]. Renewable energy sources come in a variety of forms and are widely available. The most cutting-edge renewable energy sources among them are PV and WECS, which are frequently used in all systems [3]. All different types of DG units are integrated into a hybrid power system called a standalone microgrid to fully take advantage of both their complementary and distinctive attributes. Due to integrated RES, a standalone microgrid system may experience power quality and stability problems. The intermittent climatic circumstances of RES also lead to an increase in unstable situations on the load side.

PQ issues namely disturbance, sag, harmonics, and swell, occur in standalone microgrid systems as a result of intermittent environmental changes affecting RES and load fluctuations such as non-linear loads, critical loads, and unbalanced loads [4]. By protecting the system from power quality problems, reliability and stability are improved. Filters and power bespoke devices of Flexible Alternating Current Transmission Systems (FACTS) devices are typically used to mitigate these issues [5]. FACTS devices are employed in standalone microgrids to address power quality issues, enabling system dependability and stability. To address PQ problems, series and shunt types of FACTS devices are connected in a microgrid. Static Synchronous Series Compensator (SSSC) and Dynamic Voltage Restoration (DVR) are the series devices that will be connected in series and used to correct the microgrid's voltage problems. Thyristor Controlled Reactor (TCR) and Distribution Static Compensator (DSTATCOM), which are connected to the shunt and correct and regulate voltage in the system, are the two components of the shunt FACTS device [6].

Distribution systems reactive power and unbalanced loading can be modified utilizing UPQC's well-known and cost-effective methods [7]. Managing compensators is challenging in a number of circumstances since they depend on the controller output signal to function. A number of convolutional controlling models were developed to solve these issues. In order to control the compensator, proportional integral derivative (PID) and proportional integral (PI) controllers are commonly employed [8]. After that, any power quality issues are addressed with Fuzzy Logic Controllers (FLC) [9]. It is challenging to solve the ideal problem for these devices, nevertheless. The system performs poorly when it comes to directing an integrated process and a protracted procedure, and its accuracy is also reliant on human aptitude and skill. An ideal controlling approach is put forth to address the issues with the best parameter selection in a controller [10]. These models, however, do not yield trustworthy results and are

unstable in critical conditions. Also, those models have lower reliability, higher price, lower performance, and more complex architecture.

FOPID controller has drawn the attention of academic and industrial researchers over the last ten years [11]. They are more flexible and function better than conventional PID controllers since five factors can be changed. However, tweaking the FOPID controller becomes challenging. To address these challenges in this controller, numerous design and tuning strategies have been developed. However, those models take more time, are less effective at tuning, are difficult to execute effectively, and are too sophisticated to answer problems in all situations. The brief review of literature is presented in Table 1.

## 2. Brief literature: An overview

This paper mainly focuses on mitigation of PQ issues in RES grid connected system. The following are the main contributions:

- A novel hybrid optimization-based FOPID controller was developed to improve PQ in integrated microgrid systems in order to mitigate these problems. That is both the Coati Optimization Algorithm (COA) and Osprey Optimization Algorithm (OOA) are combined to make a hybrid optimization CO-OA algorithm. Hybrid optimization reduces the time of the decision-making process and allows to focus their time on the analyses.

- Designed to increase PQ in a hybrid RES-grid connected non-linear distribution system. A hybrid RES based Grid with a load model was designed as per specific ranges, in between load and hybrid RES an UPQC is linked for PQ management.

- The pulse generation of UPQC compensator's switches is done through the use of the FOPID controller, which compares the actual value and reference value to generate the pulse signal of the switches. FOPID controller's parameters are tuned by using an innovative hybrid optimization algorithm.

- A low error function is used to choose the best optimal values. The proposed mitigation performance is analyzed under sag, harmonics, interruption, and swell conditions. Moreover, the outcomes are compared to some other existing models to confirm the efficacy of the proposed model.

The remaining portions of the manuscript are structured as follows: The general design and procedure of the proposed work are described in Section 3. The performance verification of the suggested model is shown in Section 4. The overall conclusions of the work are presented in Section 5.

## 3. Proposed methodology

Power conversion, control, and usage has to be effective due to the sensitive loads in the utility, commercial, and industrial sectors. The production of power electronic devices on a wide scale exposes the power system to harmonic injection, short-term voltage swings, and other harmful power quality problems. A dynamic custom power device known as UPQC, which is extensively utilized, is used to address PQ issues in the power system. A hybrid optimization-based FOPID controlling strategy is used to regulate this device. Fig 1(a) depicts the architecture of the suggested model for improving power quality.

A standard microgrid system with loads is included in the proposed architecture. Renewable energy sources including solar, batteries and utility grids are regarded as source side is linked to the UPQC model. To convert the source electricity, place a boost converter is used to control the solar PV power on the source side. 3 phase non-linear loads are all time energized

**Table 1. Review of literature.**

| | | |
|---|---|---|
| • Proposed UPQC to improve power quality by reducing voltage sag, voltage swell, and current harmonics on the source current.<br>• The adaptive PI controller generated reference signals for both the shunt and series APFs. Park's transformation was used as a control approach to tune the gain values to attain the voltage reference level. The adaptive PI controller can be used for both open and closed loop control. | D. Sunitha et al | [12] |
| • Designed double loop control technique and experiment design for UPQC. The simulation of a compensation approach for voltage sag and harmonic current demonstrates the usefulness of the double loop.<br>• Harmonic suppression and voltage sag compensation, as well as detailed experiments for harmonic compensation, single phase voltage drop, and two phase voltage drop compensation to validate the system's design. | M. Yang et al | [13] |
| • This research proposes a fuzzy-based incremental conductance regulated MCUPQC. It improves the power quality of grid-connected WES systems. In comparison to a traditional UPQC, the novel topology can protect sensitive and important loads from interruptions, distortions, and unbalanced signals. The MC-UPQC's performance was calculated under various disturbance circumstances. This study focuses on voltage swell/sag caused by faults, feeders, or load variations in one of the feeders.<br>• The MC-UPQC with hysteresis controller is designed to prevent voltage imbalances such swell, sag, interruption, and harmonics. Furthermore, the voltage governance was done by protecting the loads from any disturbances. | P. Chaudhary et al | [14] |
| • Designed an ASO-based FOPID with UPQC to lower THD and address issues with swell, sag, non-linear load, disturbances, and unbalanced load.<br>• This strategy is efficient even though hybrid algorithms with improved performance characteristics can be employed to control strategies | C.R. Reddy et al | [15] |
| • Proposed PV, WT, and BESS structures were set up for the HRES technique. The BESS device balances load demand in environmental conditions or when PV and WT power production is insufficient to meet grid demands.<br>• The FOPID controller was integrated with BWO to equip the DPFC with two controllers, including active and shunt active power filters. The DPFC technology helps reduce voltage and improve power efficiency. | B. S. Goud | [16] |
| • Designed UPQC optimal placement and reconfiguration method is utilized in the distribution network to reduce losses, improve the voltage profile, and increase system dependability.<br>• The two objective functions of losses and mean voltage deviation are defined using the reliability indices. Because the reconfiguration of distribution networks is a large-scale nonlinear hybrid optimization problem with several constraints, the genetic algorithm combines genes including key numbers and UPQC location, UPQC series power size, and UPQC shunt power size to solve the problem. | Masoud Dashtdar et al | [17] |
| • Employed an improved FOPID controller with a GS algorithm to demonstrate a method for DVR. This approach was successful in resolving a number of PQ issues, including fault compensation, voltage regulation, sag, THD reduction, and swell, although it has a complex structure | Rajendran et al | [18] |
| • Introduced the Cuckoo optimization method, which modifies the PI controller's parameters in shunt controllers to lower harmonic distortions and improve power quality.<br>• Instantaneous PQ theory is used to produce the reference signals necessary for shunt and series controllers together with DQ-transformation evaluation. However, this method is not suitable for the unity power factor mode of operation. | Budi et al | [19] |
| • A modified UPQC based on modular multilevel converters has been built to address supply voltage and current-related power quality challenges.<br>(i) Adjust voltage sag<br>(ii) Lower source current harmonics to 2.87%.<br>(iii) Regulate voltage at PCC for unbalanced and variable loads.<br>(iv) Limit voltage harmonics to 3.89%. | T. M. Thamizh Thentral | [20] |
| • Designed the PV-UPQC system, which was tested using a reinforced learning algorithm and an adaptive neuro-fuzzy controller.<br>• By assisting to generate reference currents and determining the system parameters using linguistic rules, the fuzzy model-based controller raises the efficiency of the system. But it was not suitable for grid connected renewable energy system. | Dheeban et al. | [21] |

using a grid system. The hybrid system contains PV and battery, which are connected to a UPQC compensator via a DC-DC converter. The three-phase system between the grid and the load is connected to UPQC. The control of UPQC is done through the use of the FOPID controller. FOPID have five various parameters which are appropriately selected to increase working performance. Consequently, the optimal problem for the FOPID controller is resolved using a hybrid optimization strategy. The Coati Optimization Algorithm (COA) and Osprey Optimization Algorithm (OOA) are hybrid to choose the optimal parameter of FOPID. The

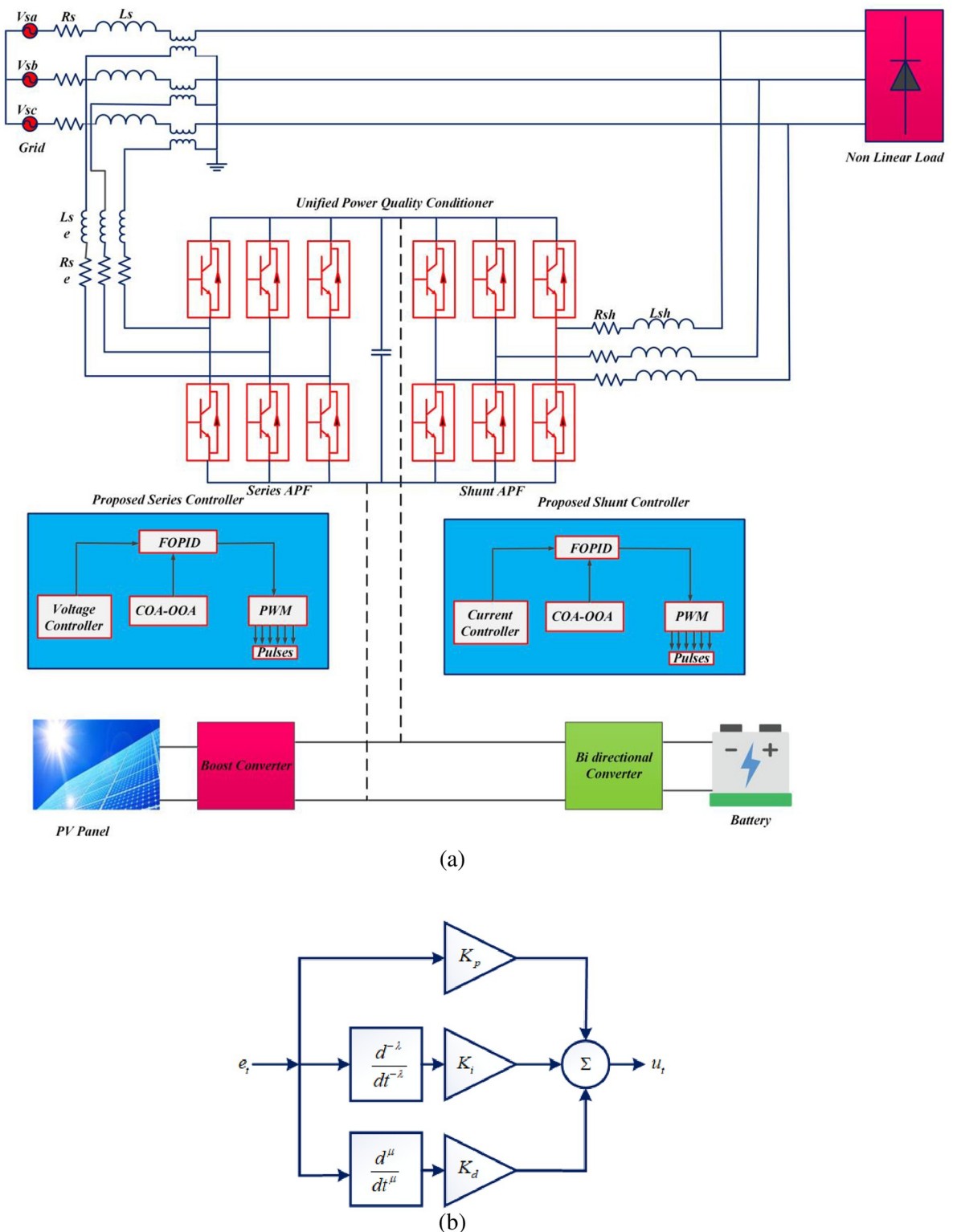

**Fig 1.** (a) Overall working process of the proposed PQ improvement model. (b) FOPID Controller.

proposed model is examined in five variety of situations, including static and normal conditions of non-linear load and static and normal condition of linear loads. In each cases, swell, interruption and sag conditions are analyzed. The proposed model's mathematical modelling is provided below.

### 3.1. Modelling of utility grid

The unit is connected between various voltage levels using transmission lines and entire transformers. Thevenin equivalent circuit coupled to a series sine wave voltage (220 V, 50 Hz) with $(Z = R + jX)$ impedance is used to develop the grid system.

$$P_{utility} = \Delta V / I \tag{1}$$

Where $I$ stands for current and $\Delta V$ stands for potential difference.

### 3.2. Modelling of PV

Solar PV systems are considered as RESs in this system. An ideal current source, a diode that replicates the $P - N$ semiconductor junction, internal shunt resistance $R_p$, and internal series resistance $R_s$ can all be used to simulate a conventional solar cell. According to the semiconductor theory, the fundamental equation for a solar cell's $I - V$ properties is provided by

$$I = I_{pv,cell} - I_{o,cell}\left[exp\left(\frac{qv}{akT}\right) - 1\right] \tag{2}$$

The general expression for solar cell current is,

$$I = I_{PV} - I_O\left[exp\left(\frac{V + R_S I}{V_t a}\right) - 1\right] - \left(\frac{V + R_s I}{R_P}\right) \tag{3}$$

Where, $I_{PV}$ is $pv$ current of the array, $I_o$ is $pv$ current of the array, $V_t$ is the thermal voltage of the array, the total number of $pvs$ is given by $N_s$, the total number of $pv$ s is given by $N_p$, $R_S$ is the equivalent series resistance, $R_p$ is the equivalent parallel resistance, and an is picked at random. Normal values are $1 \leq a \leq 1.5$ and depend on other $IV$ model parameters. A solar cell's current that generates light is,

$$I_{pv} = \left(I_{pv,n} + K_I \Delta_T\right)\frac{G}{G_n} \tag{4}$$

Where $G$ the irradiation on the device surface is, $K_I$ is the current coefficient, and $G_n$ is the nominal irradiation. $I_{pv,n}$ is the light generated current at a temperature of $25°C$ and a power density of $1000 W / m^2$, $\Delta_T$ is the difference between the nominal and real temperatures, and $I_o$ and its temperature is represented by

$$I_0 = I_{0,n}\left(\frac{T_n}{T}\right)^3 exp\left[\frac{qE_g}{ak}\left(\frac{1}{T_n} - \frac{1}{T}\right)\right] \tag{5}$$

Where the nominal saturation current $I_{0,n}$, is

$$I = \left[\frac{I_{SC,n}}{exp\left(\frac{V_{oc,n}}{aV_{t,n}} - 1\right)}\right] \tag{6}$$

Where $V_{t,n}$ the thermal voltage at the nominal temperature and $E_g$ is the semiconductor's band gap energy. At nominal temperature $T_n$, short-circuit current $I_{SC,n}$ and open-circuit voltage

$V_{oc,n}$ are measured. Solar energy is converted into DC power via the DC-DC boost converter that links the PV array to the DC bus. Indeed, because of the nonlinear characteristics of PV panels and stochastic variations in solar irradiation, a Maximum Power Point (MPP) is always available for each specific operating situation of a PV array.

### 3.3. Modelling of DC-DC boost converter

DC-DC converters are used in PV systems to regulate the voltage that the PV modules produce. The DC-DC boost converter circuit is made up of an inductor ($L$), a diode ($D$), a capacitor ($C$), a load resistor($R_L$), and a control switch ($S$). These components are connected in a way that ramps up the voltage from the input voltage source ($V_{in}$). The duty cycle of the control switch affects the boost converter's output voltage. So, changing the switch's ON time will change the output voltage. As a result, the average output voltage for the duty cycle " $D$ " may be computed using,

$$\frac{V_0}{V_{in}} = \frac{1}{(1-D)} \tag{7}$$

Where $V_{in}$, $V_0$ are the converter's input and output voltages, and $D$ the duty cycle of the control is switch, respectively. The output power of the converter is equal to the input power in an ideal circuit,

$$P_0 = P_{in} \rightarrow V_0 I_0 = V_{in} I_{in} \tag{8}$$

### 3.4. Modelling of battery

An electric vehicle (EV) battery is designed to have an internal resistance that varies in response to the load and flow of electrical power. The lithium (Li)-ion battery, which has better performance qualities like a longer lifespan, a higher energy density, and less weight, is taken into consideration in this work. As the SOC drops and the battery's current draw expands, the battery voltage tends to decrease.

$$SOC_{EVB\_min} \leq SOC_{EVB} \leq SOC_{EVB\_max} \tag{9}$$

Where, $SOC_{EVB\_min}$ and $SOC_{EVB\_max}$ are EVB's permitted minimum and maximum states, respectively.

### 3.5. Modelling of bidirectional DC-DC converter

The DC-DC converter functions as the link between the battery pack and the DC link bus in electric vehicles, allowing power to flow from high voltage to low voltage when functioning as a buck converter and from low voltage to high voltage when working as a booster. Because of this, using a bidirectional DC-DC converter offers the advantage of adaptable DC-link voltages, which can increase system efficiency and result in smaller component sizes. It also enables the use of different energy storage systems [20–25].

### 3.6. UPQC

The UPQC consists essentially of two power electronic converters connected by a single DC wire. Power electronics-based appliances for power conditioning can be used as a crucial solution to increase the power grid's efficiency. PQ issues will be addressed with this device. Several FACT devices can be used to solve the system's power quality issues. In a manner similar to

this, the UPQC-PQ can be utilized to handle issues with voltage and current PQ. Harmonics, flickers, imbalance, sag, and swell are all examples of PQ issues. The Voltage Source Inverters (VSI) that the UPQC-PQ generally uses are shunt and series Active Power Filter (APF), as well as a DC connection capacitor. The DC-link capacitor, a key element, is required to control the voltage between two filters. The power system of the UPQC-PQ device can be broken down into a variety of distinct parts, such as active filters for series and shunt circuits and generation with a power supply system. Kirchhoff's law, which is explained in (10) and (11), is the mathematical representation of the power supply system.

$$V^{if} = e^i - L^s \frac{di_s}{dt} - R^s I^{is} - V^{ih} \tag{10}$$

$$I^{is} = I^{iL} - I^{ih} \tag{11}$$

Where $I^{ih}$ is the output current of a shunt active filter, $I^{iL}$ is the load current, $I^{is}$ the line current, $V^{ih}$ is the output voltage of a series active filter, $e^i$ is the source voltage, $L^s$ is the inductance of the transmission line, $R^s$ is the resistance of the transmission line, and $i$ in the subscript is described as the a, b and c phases in a power system. A series and shunt active filter controls the source current and load voltage in this UPQC-PQ system.

The series and shunt active filters supply the proper voltage and current when issues with power quality arise in the power system. $I(ch)$ and $I(s)$, which represent the load and source currents, respectively, are frequently utilized. $I(f)$ reflects an active power filter's injected current, while $V(c)$ shows an active power filter's injected voltage. The series active filter's injected voltage serves as the reference load voltage, indicated by $V(ch)$. The source voltage factor fluctuation is denoted by the symbol $k$, while the load power factor is specified as $\cos\varphi(n)$. Eq (12) displays the percentage difference between the source voltage and the reference voltage. When the system has reached over voltage $\acute{V}_G$, the series inverter injects the negative voltage $V_{series\,inverter}'$ into the grid to remove it [26–29].

$$V(c) = V(ch) - V(s) = -kV(ch) < 0^\circ \tag{12}$$

Eq (18) can be used to resolve the equation above.

$$k = \frac{V(s) - V(ch)}{V(ch)} \tag{13}$$

The losses are omitted in UPQC-PQ designs. Active power and load power requirements are compared to PCC input power requirements. As seen in Eq (14) below, the PCC side current is,

$$I(f) = \frac{I(ch)}{1+k} \cos\varphi(n) \tag{14}$$

Even if in the steady state the average dc link voltage stays at a predetermined level, the UPQC dc link voltage can deviate significantly from its reference during transient events brought on by load connection/disconnection or/and supply side voltage sag/swell. Due to the high dc link voltage fluctuation, the magnitude of the series injected voltage cannot remain constant during such transients, which influences the load voltage magnitude, which changes.

**3.6.1. FOPID controller.** FOPID controller is advanced controller has two additional factors of the integrator and the order of the differentiator provide further opportunities for controller improvement in addition to $K_P, K_I,$ and $K_D$. The fractional-order differential Eq (15)

**Table 2. Controller parameters I mean gains.**

| Parameter | Ranges |
|:---:|:---:|
| $\mu$ | 0.0455 |
| $K_D$ | 0.0427 |
| $K_I$ | 2.9161 |
| $K_P$ | 0.0284 |
| $\lambda$ | 0.0313 |

provides a description of the FOPID controller.

$$u(t) = K_P e(t) + K_I D^{-\lambda} e(t) + K_D D^{\mu} e(t) \tag{15}$$

where $\mu$ signifies differential order, $\lambda$ denotes integral order, both $\mu$ and $\lambda$ are positive real numbers, $K_D$ represent differential gain, $K_P$ signifies proportional gain and $K_I$ states integral gain. The advantage of using this controller is that it performs better and is more flexible than a regular controller. The FOPID controller's tuning was still gets challenging. Select the optimal value for the controller parameters like $\mu$, $\lambda$, $K_D$, $K_I$, $K_P$ to provide a steady and reliable operation. The hybrid optimization approach was used to discover a solution to the FOPID's optimal problem. That is Coati Optimization Algorithm (COA) is utilized for the initialization process and the Osprey Optimization Algorithm (OOA) is utilized for the initialization process. Table 2 represent controller parameters. Fig 1(b) represents FOPID controller diagram. The background of the Hybrid optimization algorithm and its optimal parameter selection process of FOPID is discussed as follows.

**3.6.2 Hybrid optimization algorithm.** *(a) Background of COA.* A COA is a newly proposed meta-heuristics algorithm which simulates the characteristics of coatis found in nature. COA's basic essence is to replicate two key behaviours of coatis: their approach when attacking and hunting iguanas and their evasion from hunters. Coatis, commonly called coatimundis, are omnivorous mammals that consume both invertebrate and small vertebrate prey. Notably, the green iguana constitutes a significant component of the coatis diet. Due to their arboreal nature, coatis frequently forage for iguanas in trees and often hunt collectively. The coatis hunting strategy may involve some group members climbing trees to startle the iguana into leaping to the ground while others rapidly attack it. However, despite their effective predation tactics, coatis are vulnerable to attacks from hunters and large raptors. The COA algorithm aims to simulate the coatis behaviors. OOA's design was inspired by a simulation of osprey activity in the wild [30–37].

*(b) Background of OOA.* The osprey is a piscivorous bird, meaning that fish makes up most of its food. It regularly catches live fish that are 25–35 cm long and weigh between 150 and 300 g. However, it can capture any fish weighing between 50 g and 2 kg. Ospreys can locate anything underwater thanks to their superb vision. When it is hovering between 10 and 40 meters above the water's surface, the osprey can spot fish below. It dips its feet into the water as it approaches the fish, then dives beneath to grab it. The osprey captures its victim, then moves it to a nearby rock where it starts to consume it. A sophisticated natural behavior that can serve as the inspiration for creating a novel optimization algorithm is the osprey's approach to fishing and carrying its catch to a suitable location for consumption [23].

**3.6.3 Steps involved for selecting optimal values in the FOPID controller.** FOPID contain five parameter that are optimally selected to find the best solution for the controller. The proposed hybrid based optimal solution of the FOPID controller is described step-by-step as follows:

Step 1: Initialization

In the COA optimization algorithm, the first stage is to initialize a set of random solutions, from which the global solution, the best solution is observed. In the proposed model, initialize the parameter of the FOPID controller as an input of the total population in COA.

$$P = \left\{ K_p,\ K_i,\ K_d, S^{\lambda},\ S^u \right\} \tag{16}$$

Where $P$ denotes the population of the problem that varies from 1 to n.

Step 2: Fitness function

In order to discover a solution to a given problem that is close to the desired problem's ideal solution, an objective function known as a fitness function is used. Here, the fitness is considered as mean square Error (MSE). Smaller error is closer to the controller system desired values.

$$Fitness = min(J) \tag{17}$$

$$J = \sum_{i=1}^{n} |y_r(t) - P_n| \tag{18}$$

Where, $y_r(t)$ denotes the reference value, $P_n$ denotes the observed value and $J$ represent MSE.

Step 3: Updation

Update until it reaches the optimal value, update each iteration value according to the osprey random position. By using the Eq (17), the solutions can be updated.

$$X_i = \begin{cases} X_i^{P1},\ F_i^{P1} < F_i; \\ X_i, else, \end{cases} \tag{19}$$

Where according to the first phase of OOA, $X_i^{P1}$ indicates the *ith* osprey's new position, $F_i^{P1}$ denote its objective function value.

Step 4: Termination

The process goes to terminate after getting the best solution which is suitable for reducing the power quality issues at all conditions. Fig 2 denotes the overall process of optimal parameter selection.

This hybrid approach to optimization solves the ideal problem for the FOPID controller. The suggested model is tested in five different scenarios, including static and normal conditions for linear loads and non-linear loads. Swell, interruption, and sag circumstances are examined in each situation. Table 3 presents hybrid optimization parameters.

## 4. Result and discussion

A hybrid optimization based FOPID controller for UPQC was introduced to mitigate PQ issues in a hybrid RES system. Both current and voltage variations are controlled through the use of UPQC. The ON/OFF pulse of UPQC is managed by using the FOPID controller. An ideal value of FOPID was selected through the use of a hybrid controller. UPQC have the capacity to manage current and voltage variation, two FOPID controller was individually designed to manage the current and voltage controller. The current controller have an error value of PCC current and reference current to make a pulse signal of shunt switches. Same as in voltage controller have an error value of PCC voltage and reference voltage to make a pulse signal of series switches. The proposed model offers a mitigation performance effectively at various load conditions like linear and non-linear load. The proposed model parameter and specification are mentioned below in Table 4.

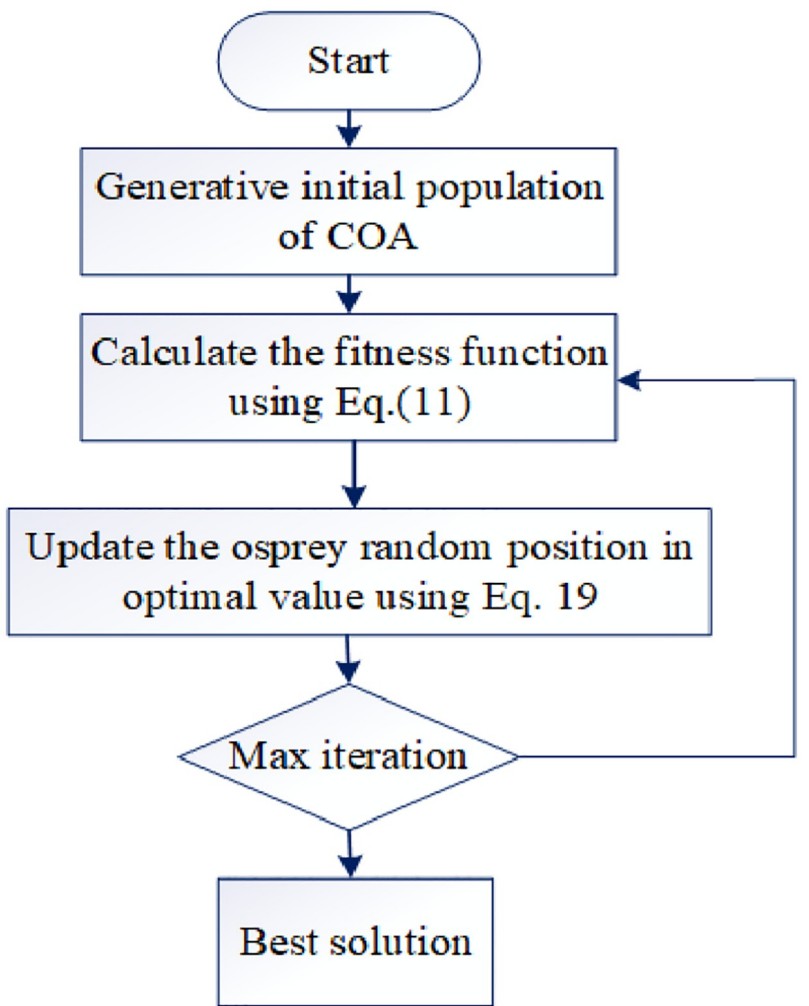

**Fig 2. Flowchart of hybrid optimization.**

The performance effectiveness of the suggested hybrid optimization-based controller was investigated using a three-phase distribution system. The outstanding performance of FOPID on the suggested controller was shown using five test studies. The analysis of the proposed system under various condition of irradiance and constant temperature were carried out. Numerous combinations of linear and non-linear loads, supply voltages and variables were used in these tests. In the section below, conditions such as irradiation, temperature, swell, sag, and disturbance are described as distinct examples.

**Table 3. Hybrid optimization parameters.**

| Parameter | Ranges |
|---|---|
| Iteration | 100 |
| Dimension | 5 |
| T | 0.01 |
| Search agent | 30 |

**Table 4. Proposed model parameter and specification.**

| Parameter | Specification |
|---|---|
| DC link voltage | 700-800V |
| Linear load | 4kw |
| Non-linear load | 6kw |
| Battery | 4 kW |
| PV | 214 W/panel<br>[I = 5.9A, V = 60V] |
| Irradiance | $1000 w/m^2$ |
| Temperature | 25˚ |
| Grid | 4.375 kW<br>[I = 35A, V = 125V] |

**Case 1**: **Normal condition with linear load**

In normal conditions solar irradiance is considered as $1000 w/m^2$ and temperature 25˚ integrated into the proposed optimal controller in linear load is shown in Fig 3. Performance analysis is done in the proposed model is explained in the section below.

In Fig 4 clearly shows the battery vs time in the graphical representation in the case 1 condition. According to SOC, the battery statement has to be followed by the battery management system. The battery's SOC is calculated as a combination of voltage and energy consumption. In Fig 4, the SOC decline of 45% is depicted as a straight line, while the voltage is shown to be nearly constant over the range's maximum of 44.995%.

Fig 5(a) depicts the grid current versus time graph. Grid current consistently maintains at 35A for 0 to 1 seconds. Fig 5(b) shows the grid voltage vs time graph. Grid voltage consistently maintains 125 volts in 0 to 1 seconds.

In Fig 6(a) depicts the PV current vs time graph. PV current 0 to 1 seconds while maintaining a constant 5.9A. In Fig 6(b) depicts the PV voltage vs time graph. PV voltage varies 0 to 0.1s and is constantly maintained in 0.1 s to 1s in 60V.

**(a) Sag fault.** The current and voltage drop below their constant levels during the sag period, typically as a result of a systemic fault. Fig 7 illustrates how power is injected during that time to balance off the load demand due to sag problems with the generated current and voltage. In other words, although the supply current and voltage are flowing constantly and without interruption, there is a sag that arises each 0.25 to 0.5 seconds. It is based on an intelligent controller.

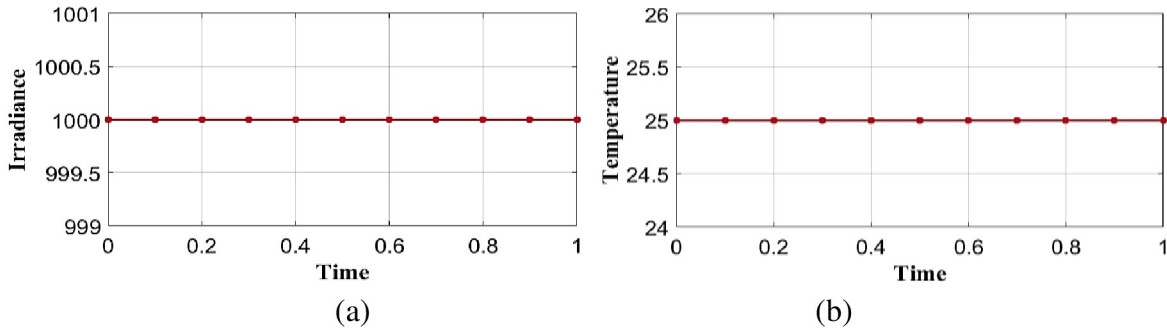

(a)                                        (b)

**Fig 3. Analysis of (a) solar irradiance (b) temperature with respect to time in case 1.**

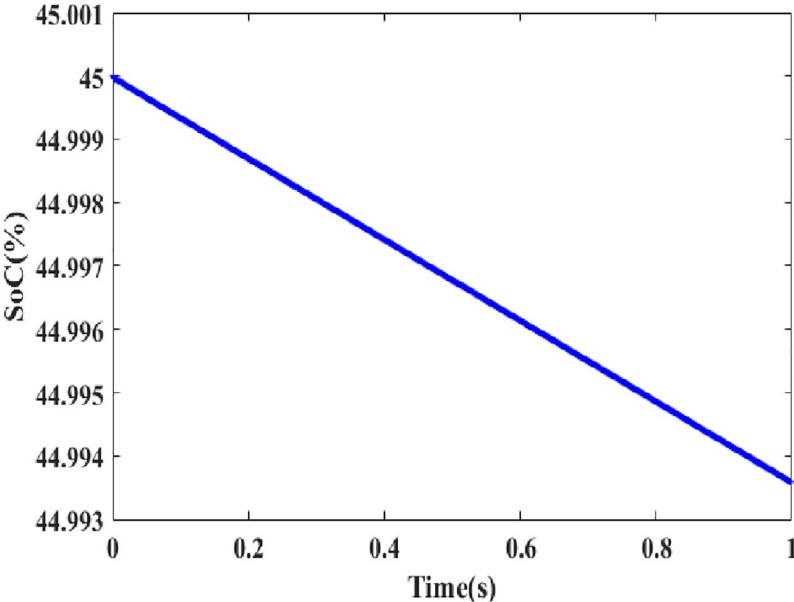

**Fig 4. Analysis of battery SOC.**

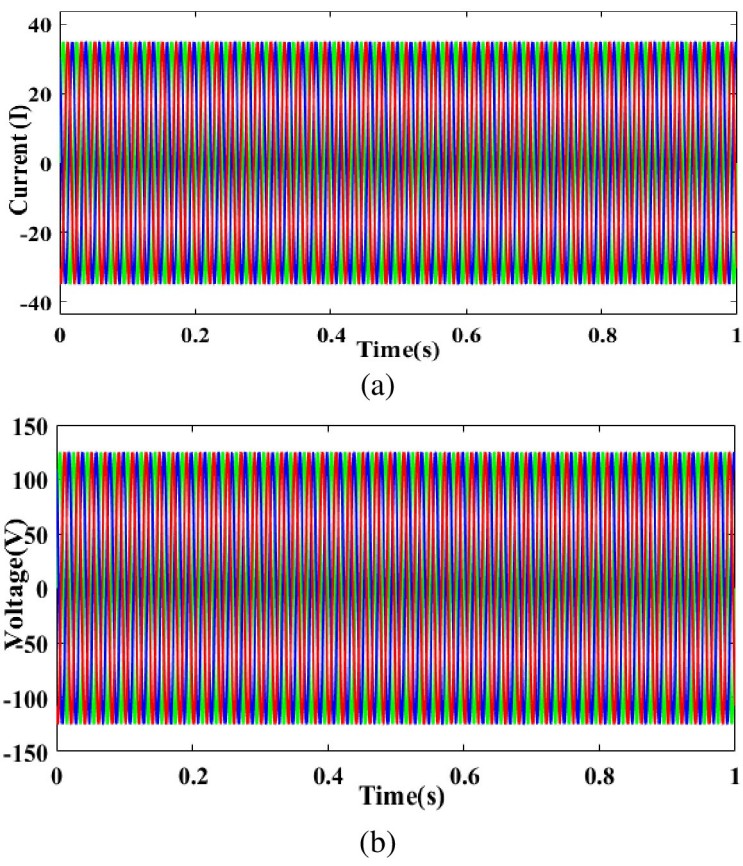

**Fig 5. Evaluation of (a) grid current and (b) grid voltage with respect to time.**

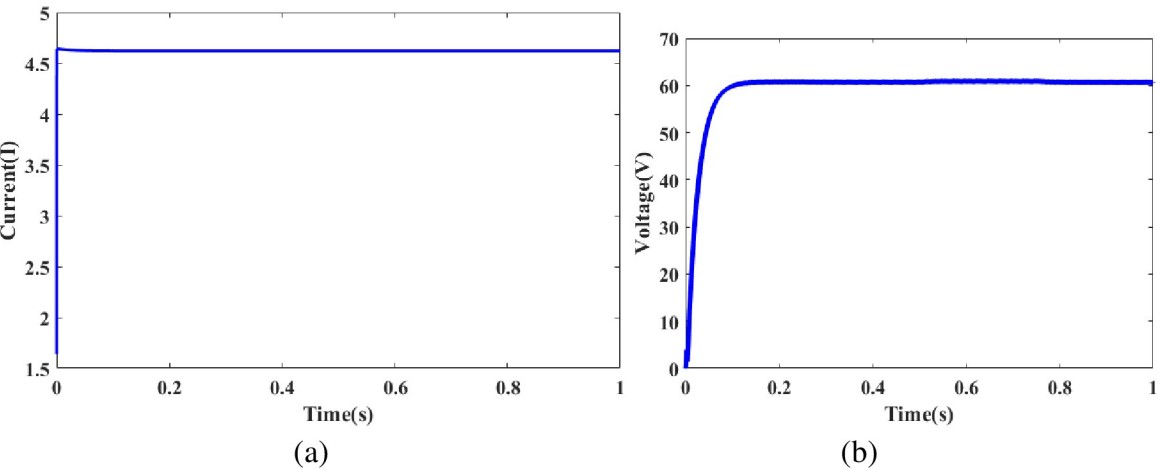

**Fig 6. Evaluation of (a) PV current and (b) PV voltage.**

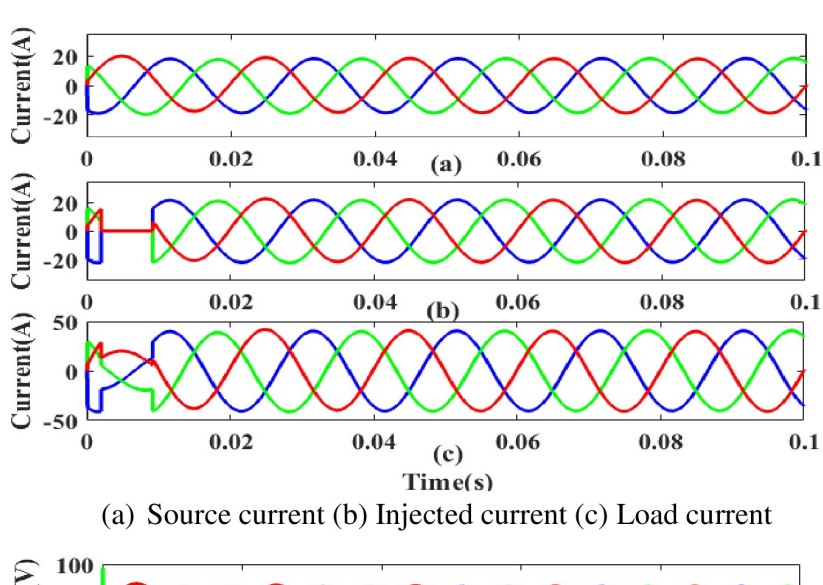

(a)  Source current (b) Injected current (c) Load current

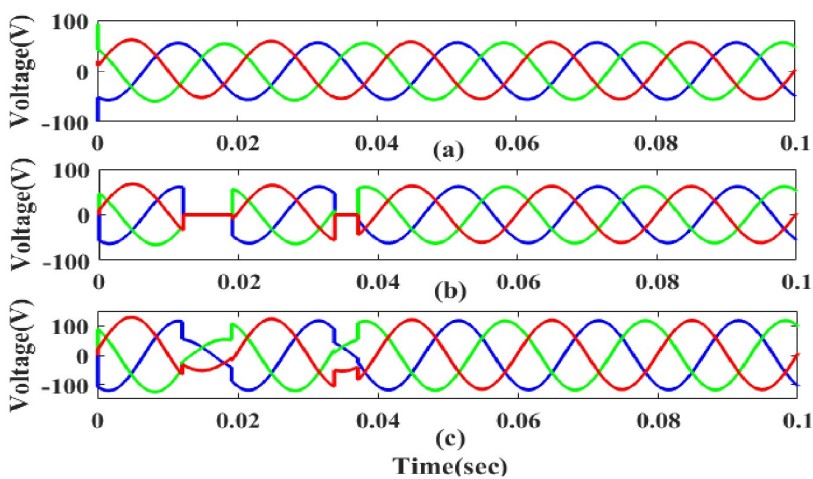

(a)  Source voltage (b) Injected voltage (c) Load voltage

**Fig 7. Evaluation of current, voltage compensation at sag period.** (a) Source current (b) Injected current (c) Load current. (a) Source voltage (b) Injected voltage (c) Load voltage.

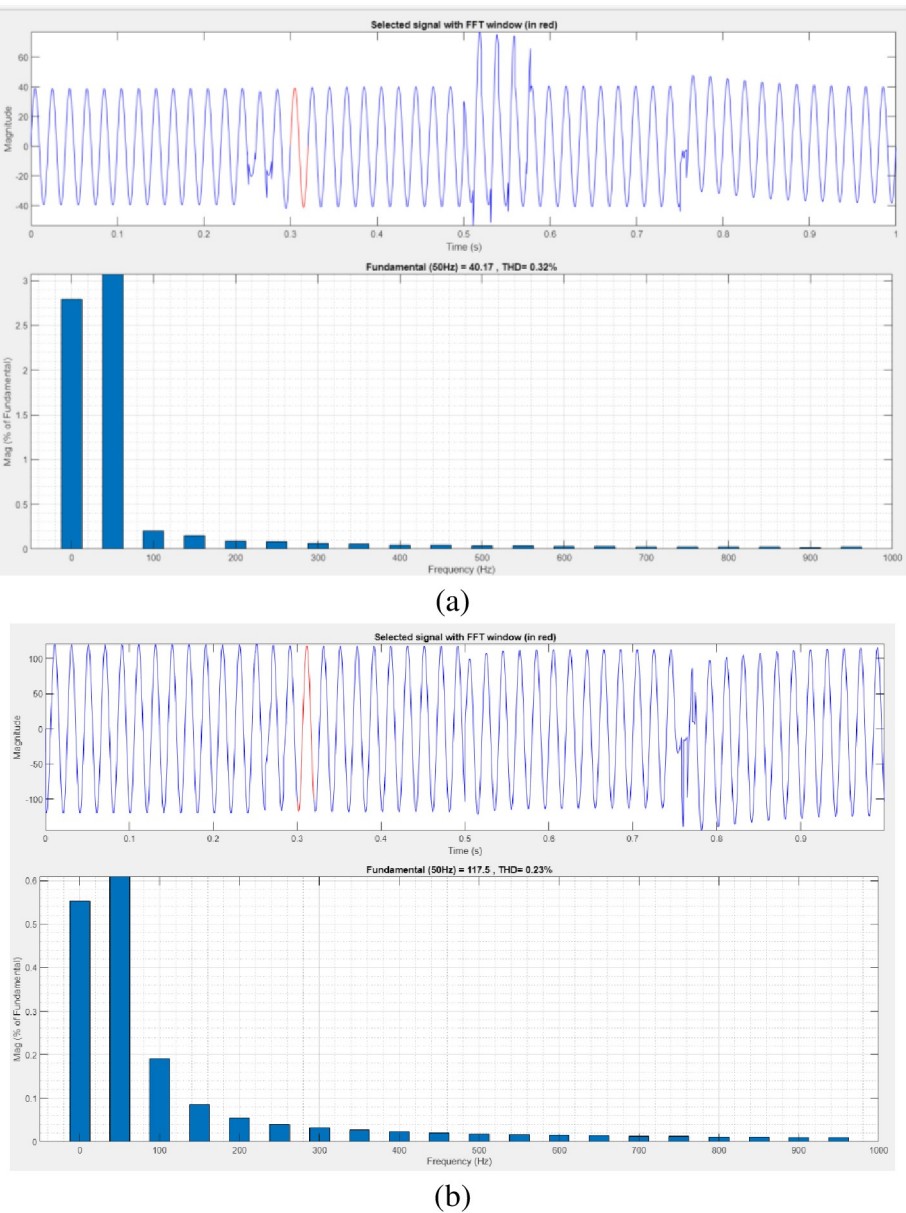

**Fig 8. THD analysis in (a) current and (b) voltage in sag condition.**

Fig 8 illustrates an analysis of total harmonic distortion (THD). Harmonics in a system can occur as a result of frequency changes. The proposed controller-based model's load current and load voltage THD values are 0.32% and 0.23%, respectively.

**(b) Swell fault.** A major issue at the load side results from swell, a low-quality power scenario in which power is increased over the level of constant voltage. Fig 9 displays the performance of the proposed controller based UPQC configuration under swell circumstances in case 1. Current is injected during that period to balance out the load demand because the generated current has swell issues, as shown in Fig 9. In other words, although the supply current is flowing constantly and without interruption, there is a swell that arises each 0. 5 to 0.75 seconds. It is based on an intelligent controller.

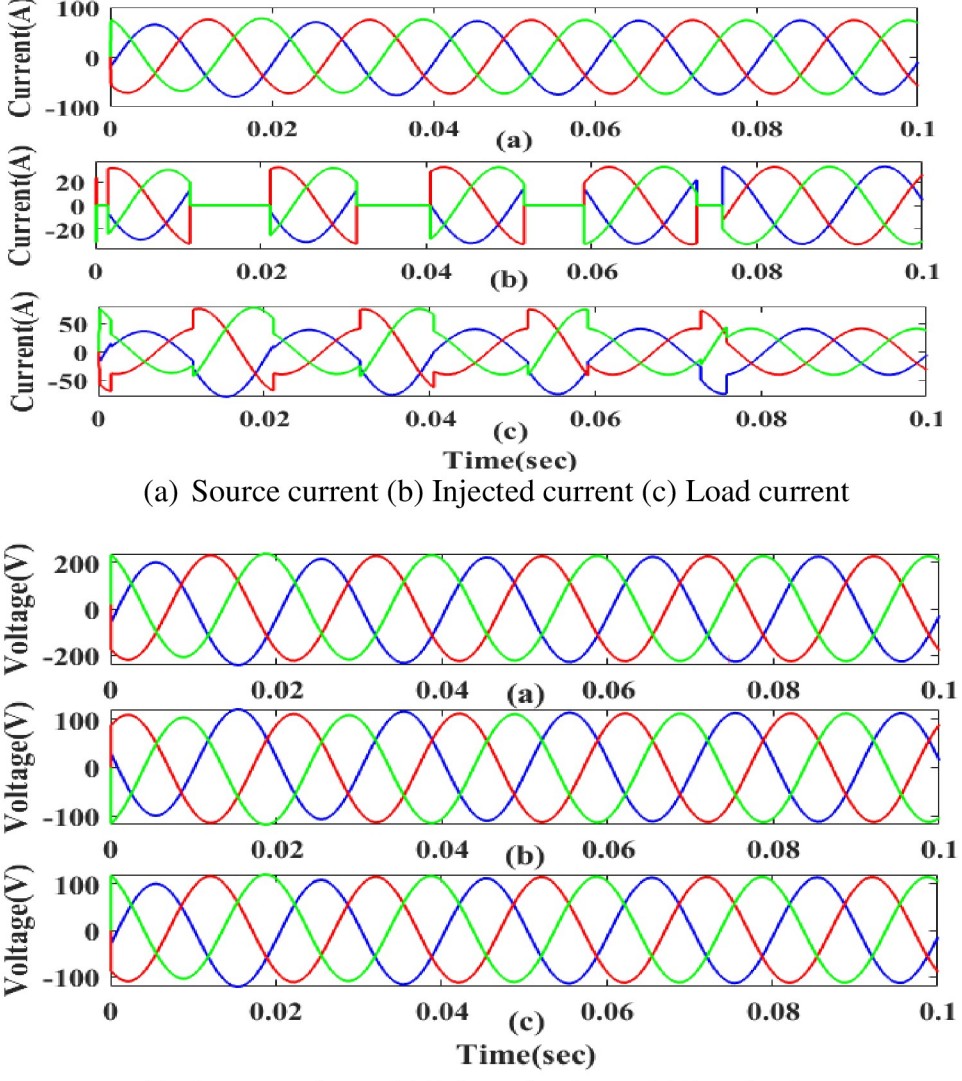

(a) Source current (b) Injected current (c) Load current

(a) Source voltage (b) Injected voltage (c) Load voltage

**Fig 9. Evaluation of current, Voltage compensation at swell period.** (a) Source current (b) Injected current (c) Load current. (a) Source voltage (b) Injected voltage (c) Load voltage.

Fig 10 illustrates an analysis of THD, harmonics in a system can occur as a result of frequency changes. The proposed controller-based model's load current and load voltage of THD is 0.02% and 0.71%, respectively.

**(c) Interruption fault.** Due to its effect on the components on the load side, interruption is also recognized as one of the system's power quality problems. Fig 11 shows the interrupted condition, the injected current and voltage, as well as the adjusted load current and load voltage. It demonstrates the interruption occurs to interrupt the power flow for 0.75 to 1 seconds based on the intelligent controller.

Fig 12 illustrates an analysis of THD, harmonics in a system can occur as a result of frequency changes. The load current and load voltage of the proposed controller-based model is 0.98% and 0.69%, respectively.

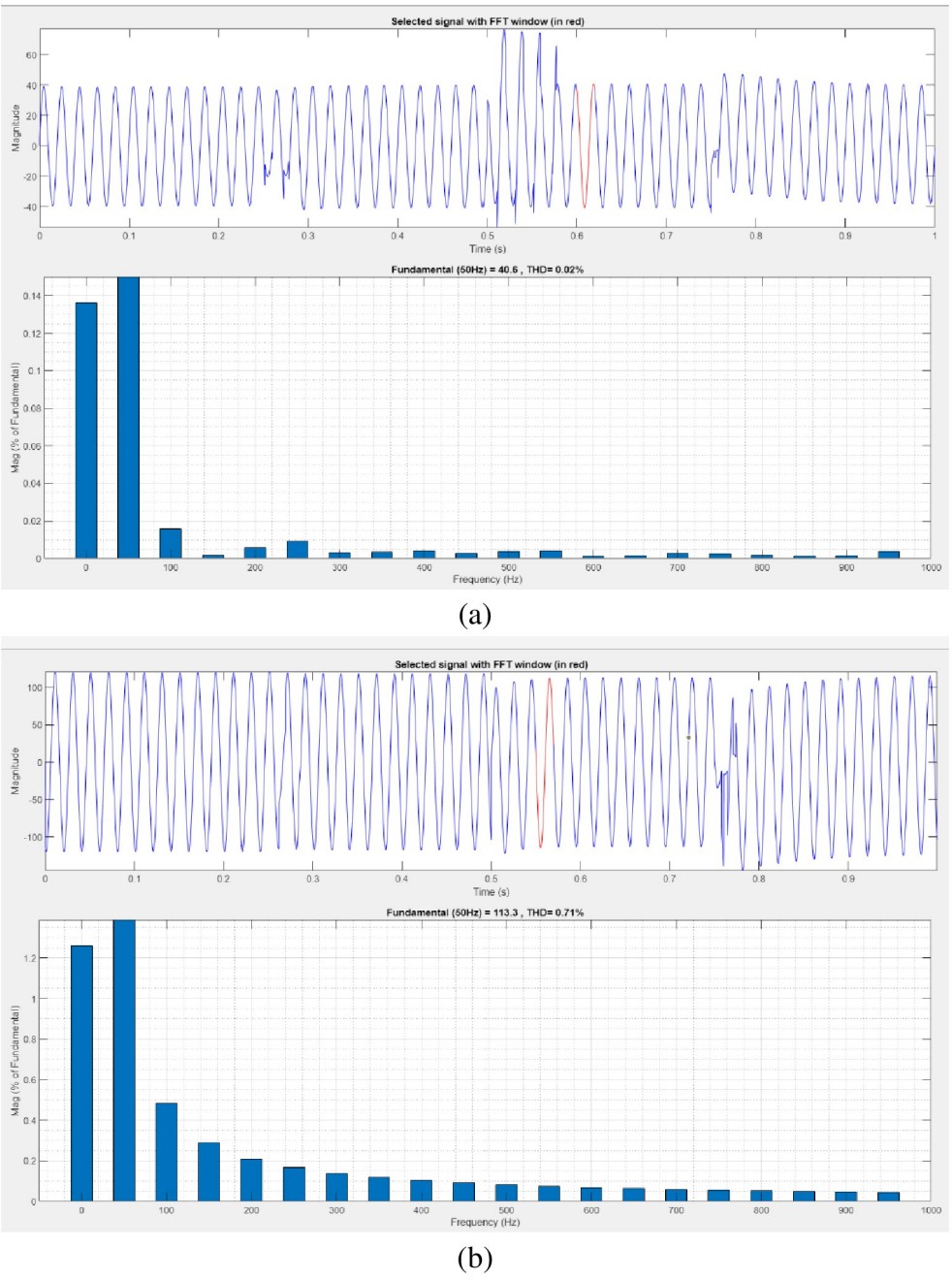

**Fig 10. THD analysis in (a) current and (b) voltage in swell condition.**

**(d) Combined fault.** Fig 13 clearly shows the combined fault that happened in current and voltage. 3phase fault voltage and current interrupt the power flow in 0. 5 to 0.75 seconds based on the intelligent controller.

**Case 2**: **Static condition with linear load**

A static shaded module generates power at a substantially lower rate than an unshaded module. In static shading conditions, solar irradiance is considered as $800w/m^2$ and

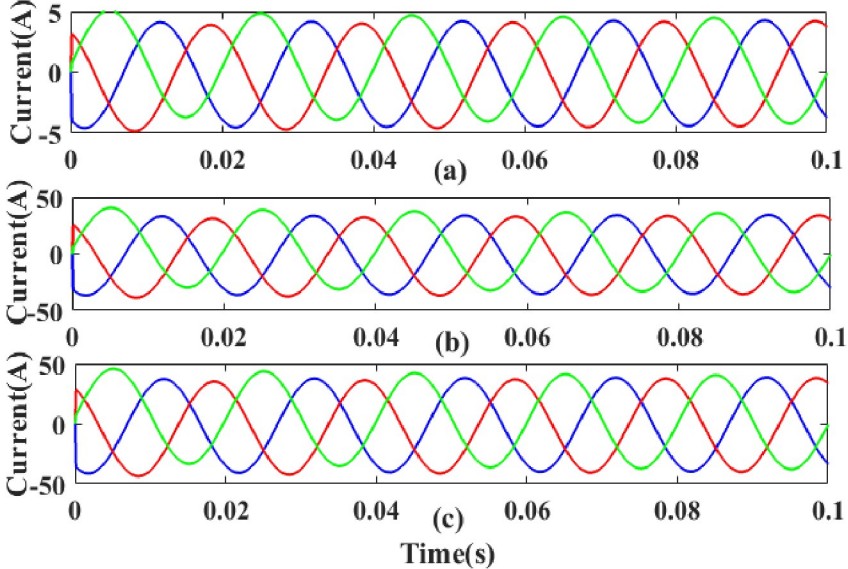

(a)  Source current (b) Injected current (c) Load current

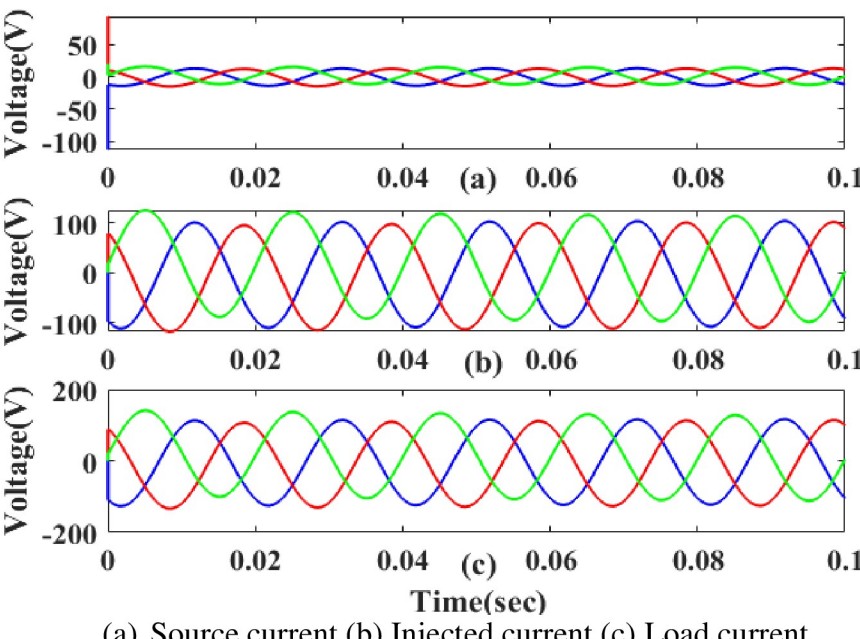

(a)  Source current (b) Injected current (c) Load current

**Fig 11. Evaluation of current, voltage compensation at interruption period.** (a) Source current (b) Injected current (c) Load current. (a) Source current (b) Injected current (c) Load current.

temperature 25° integrated into the proposed optimal controller in linear load is shown in Fig 14. Performance analysis is done in the proposed model is explained in the below section.

**(a) Sag fault.** During the sag period in static condition in linear load, the current and voltage decreases below its constant level, usually as a result of a systemic fault. The generated current has sag concerns, as shown in Fig 15. Hence, power is injected during that time to balance off the load demand. In other words, although the supply current and voltage are flowing

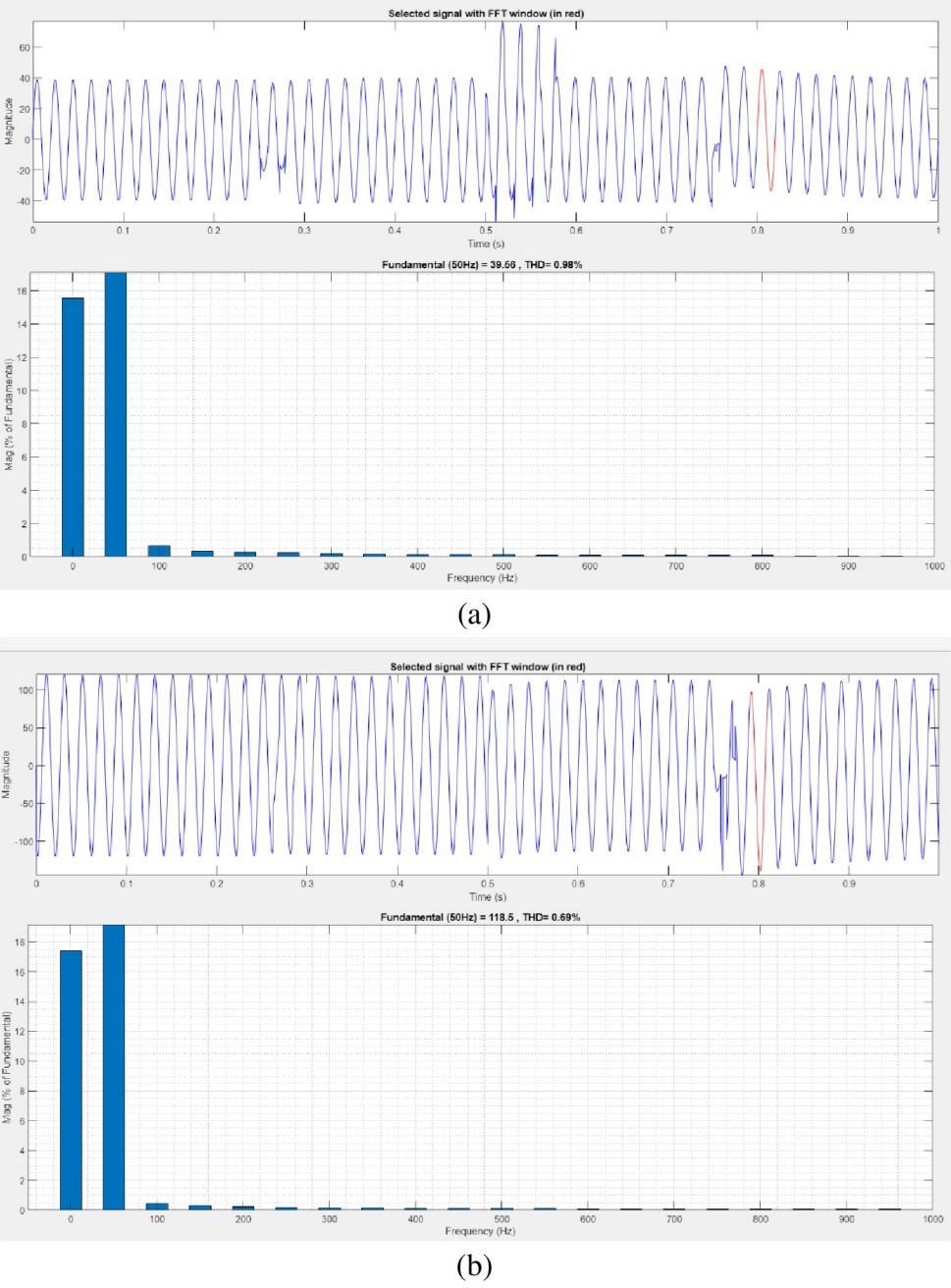

**Fig 12. THD analysis in (a) current and (b) voltage in interruption condition.**

constantly without interruption, there is a sag that arises each current and voltage is 0.25 to 0.5 seconds. It is based on an intelligent controller.

Fig 16 illustrates an analysis of total harmonic distortion (THD). Harmonics in a system can occur as a result of frequency changes. The proposed controller-based model's load current and load voltage THD values are 0.11% and 0.36%, respectively.

**(b) Swell fault.** The power that is increased above the level of constant voltage results in swell, a condition of poor PQ significantly affects the load side. Fig 17 shows the effectiveness

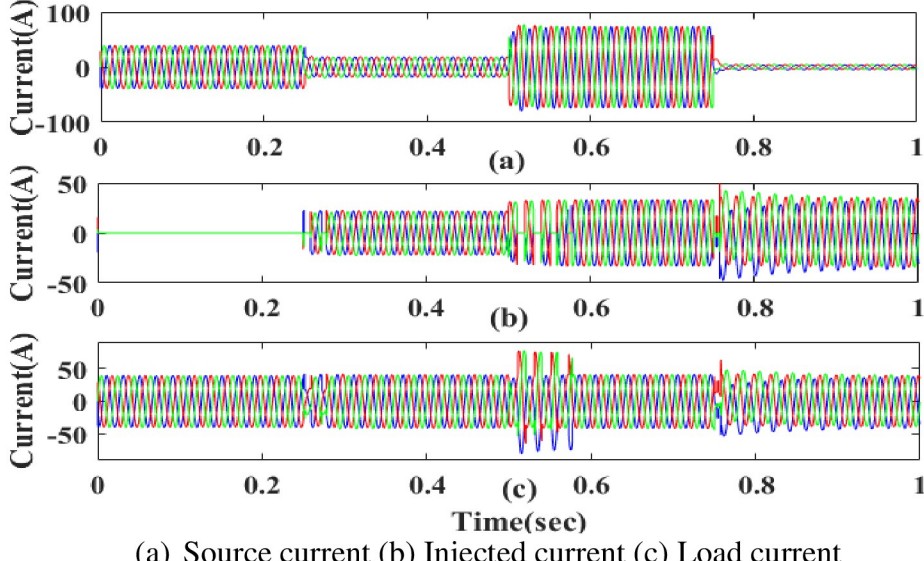

(a) Source current (b) Injected current (c) Load current

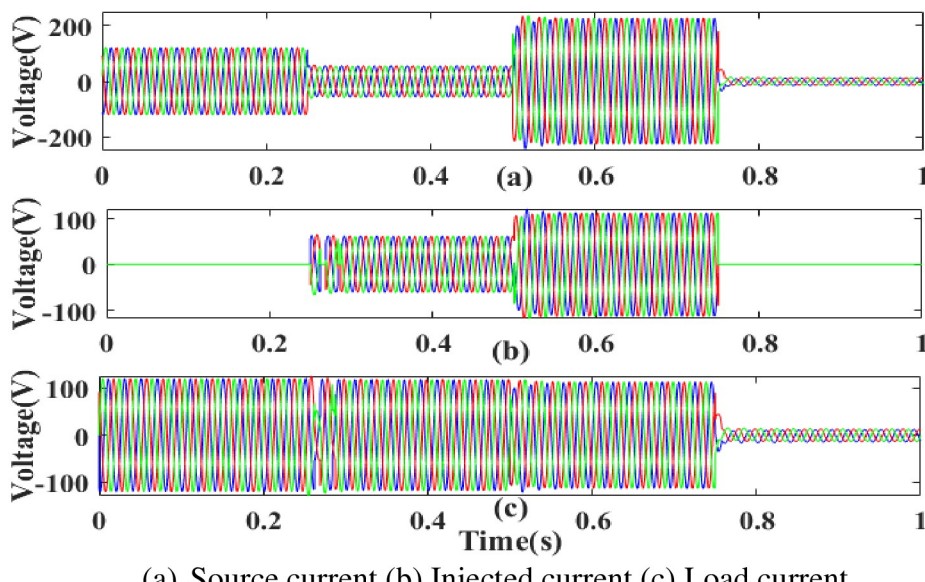

(a) Source current (b) Injected current (c) Load current

**Fig 13. Evaluation of current and voltage compensation at combined fault condition.** Source current (b) Injected current (c) Load current. (a) Source current (b) Injected current (c) Load current.

of the suggested controller-based UPQC configuration under scenario 2's swell conditions. Current and voltage are injected during that period to balance out the load demand because the generated current has swell issues, as shown in Fig 17. In other words, although the supply current and voltage are flowing constantly and without interruption, there is a swell that arises each current and voltage is 0. 5 to 0.75 seconds. It is based on an intelligent controller.

Fig 18 illustrates an analysis of THD harmonics in a system that can occur as a result of frequency changes. The proposed controller-based model's load current and load voltage THD values are 0.04% and 0.71%, respectively.

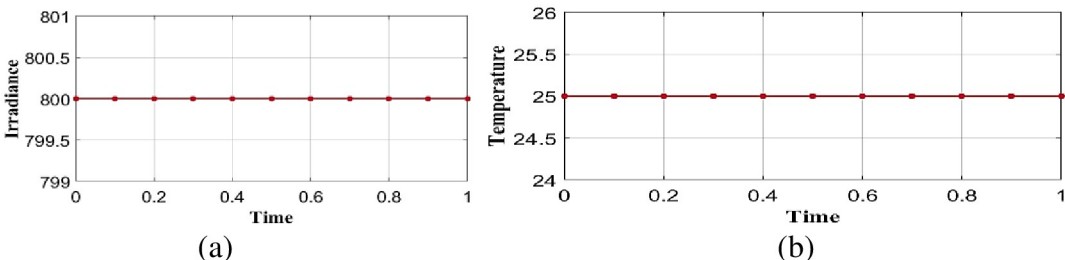

**Fig 14. Analysis of (a) solar irradiance (b) temperature in case 2.**

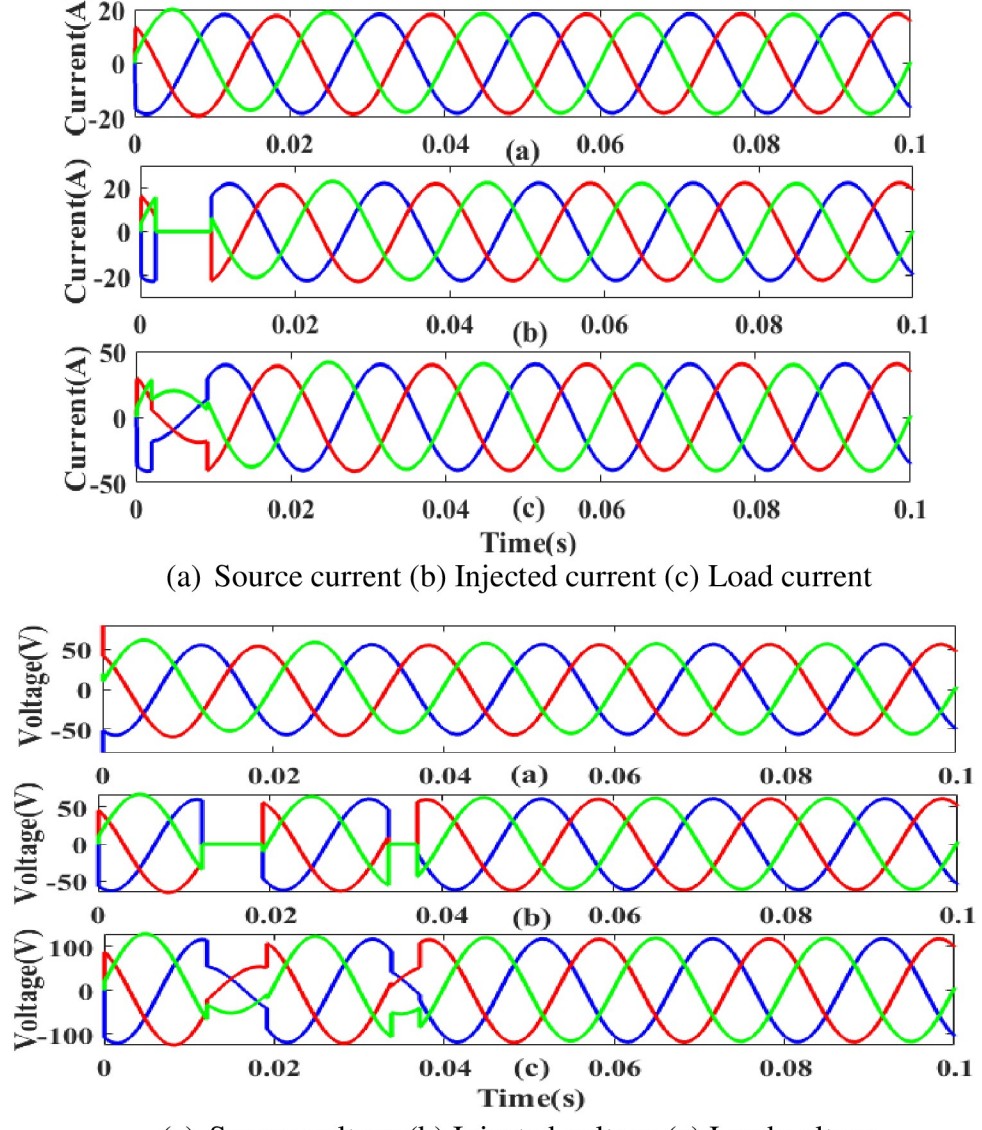

(a) Source current (b) Injected current (c) Load current

(a) Source voltage (b) Injected voltage (c) Load voltage

**Fig 15. Evaluation of current compensation and voltage compensation at sag period.** (a) Source current (b) Injected current (c) Load current. (a) Source voltage (b) Injected voltage (c) Load voltage.

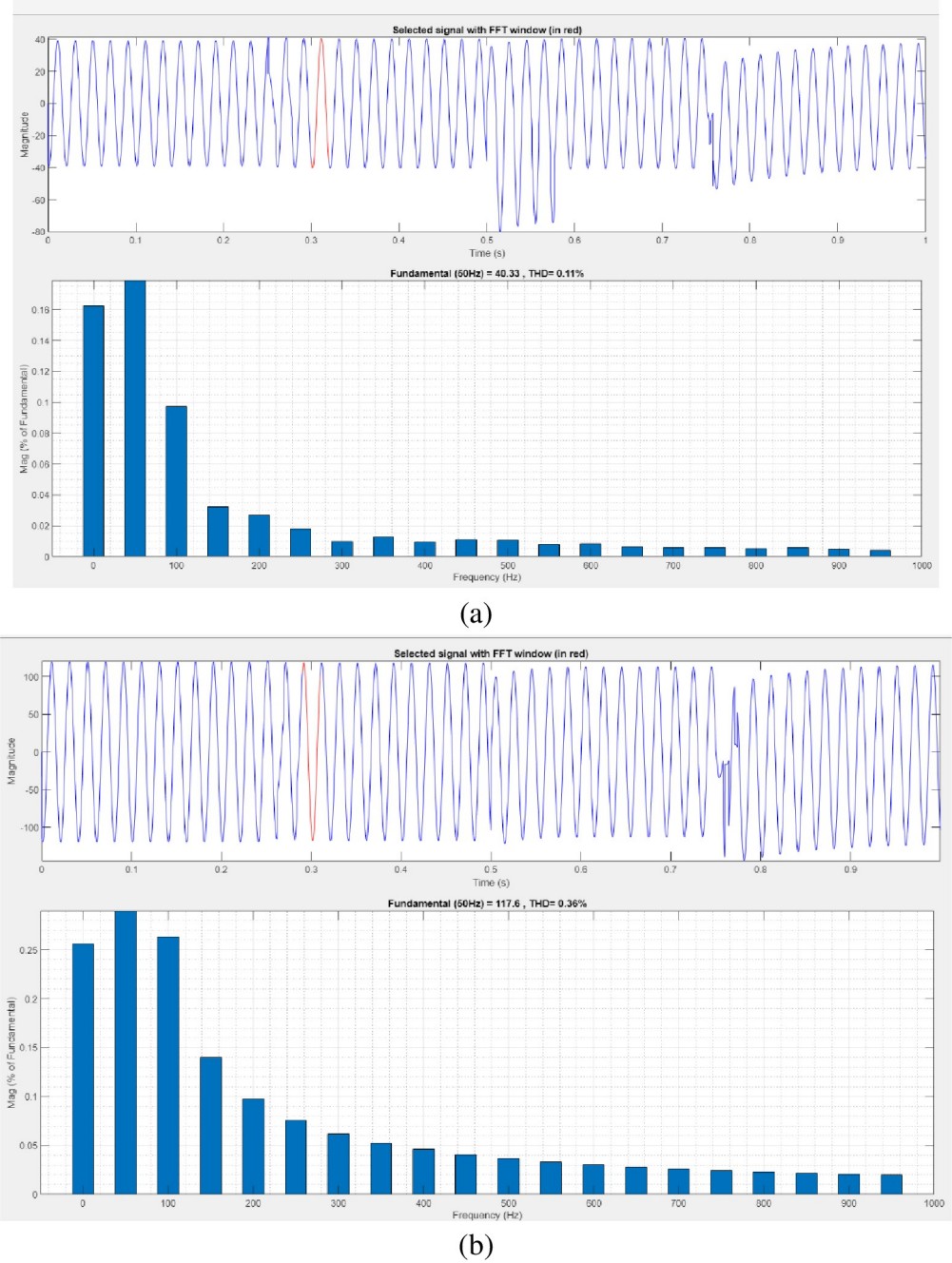

**Fig 16. THD analysis in (a) current and (b) voltage in sag condition.**

**(c) Interruption fault.** Interruption is one of the PQ concerns with the system and it affects the load side components. Fig 19 shows the interruption status as well as the injected current and voltage as well as the rectified load current and load voltage. It demonstrates the interruption occurs to interrupt the power flow for 0.75 to 1 seconds based on the intelligent controller.

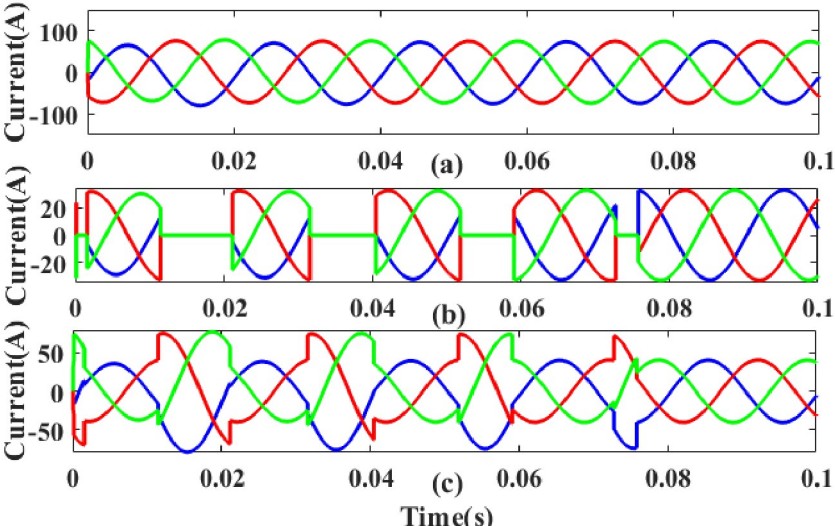

(a) Source current (b) Injected current (c) Load current

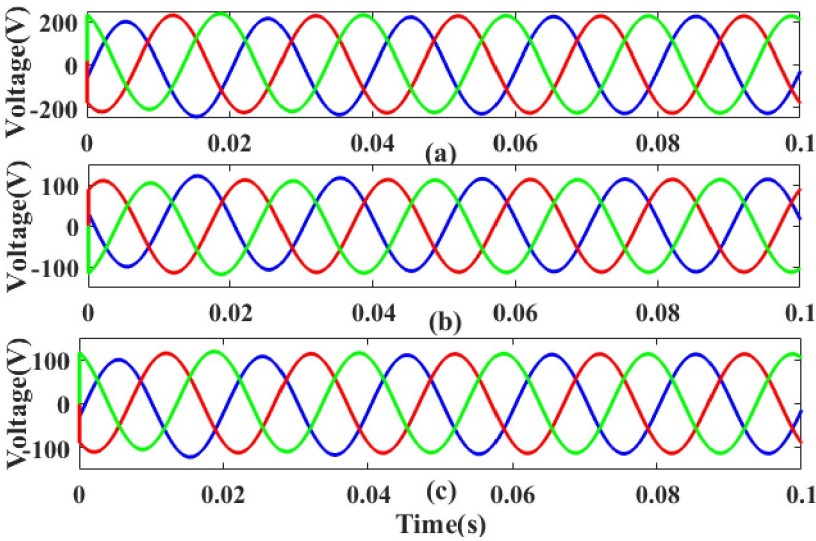

(a) Source voltage (b) Injected voltage (c) Load voltage

**Fig 17. Evaluation of current compensation and voltage compensation at swell period.** (a) Source current (b) Injected current (c) Load current. (a) Source voltage (b) Injected voltage (c) Load voltage.

Fig 20 illustrates an analysis of THD harmonics in a system that can occur as a result of frequency changes. The proposed controller-based model's load current and load voltage of THD is, respectively, 1.18% and 1.10%.

**(d) Combined fault.** Fig 21 clearly shows the combined fault in current and voltage. 3 phase fault voltage and current interrupt the power flow in 0. 5 to 0.75 seconds based on the intelligent controller.

**Case 3**: **Normal condition in nonlinear load**

In case 3 condition, solar irradiance is considered as $1000w/m^2$ and temperature 25° integrated into the proposed optimal controller in nonlinear load is shown in Fig 23. Performance analysis is done in the proposed model is explained in the below section.

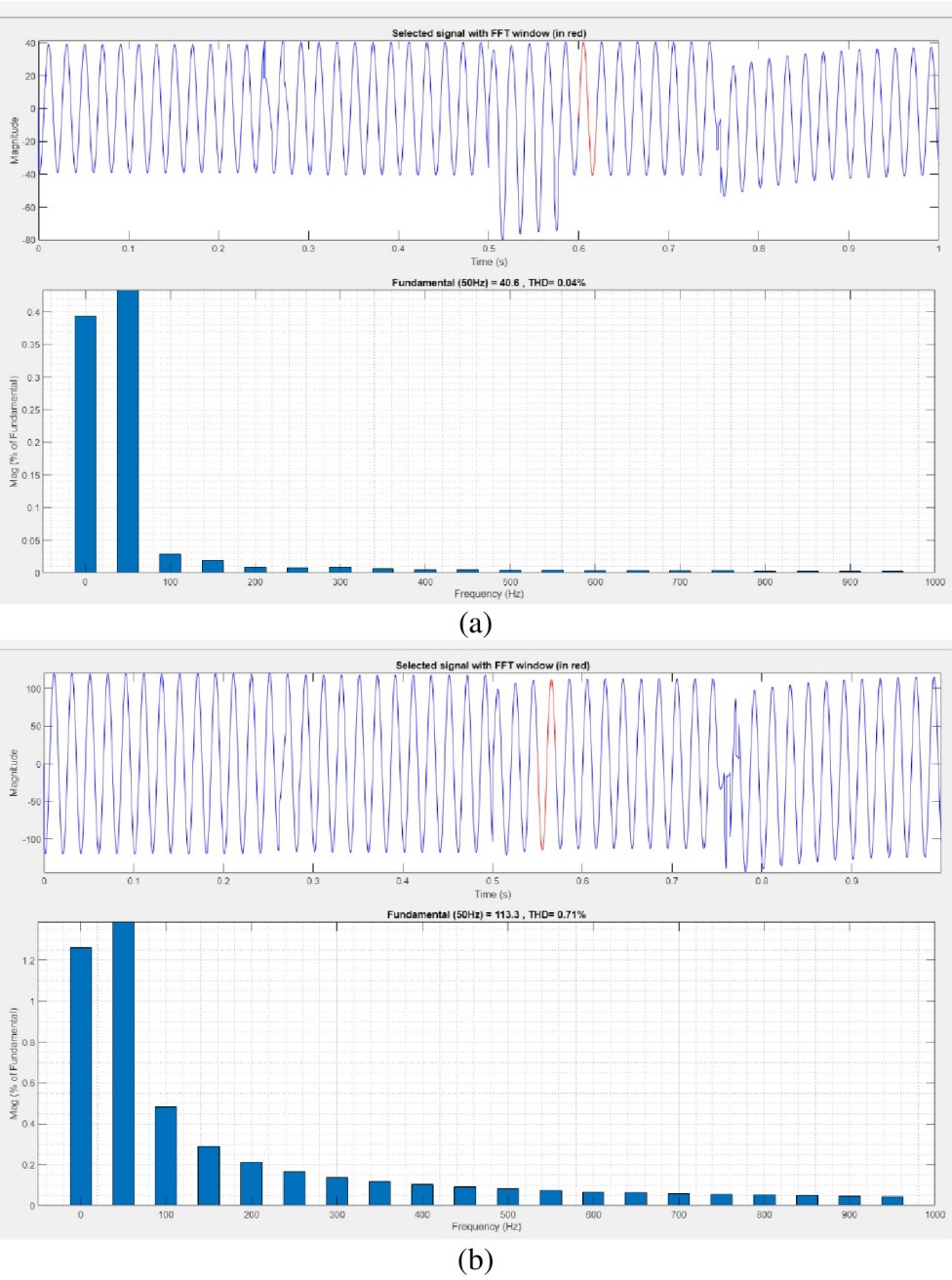

**Fig 18. THD analysis in (a) current and (b) voltage in swell condition.**

**(a) Sag fault.** As seen in Fig 22, the generated current has sag concerns. Hence, power is generated during those times to balance out the load demand. In other words, although the supply current and voltage are flowing constantly without interruption, there is a sag that arises each current and voltage is 0.25 to 0.5 seconds. It is based on an intelligent controller.

Fig 23 illustrates an analysis of THD harmonics in a system that can occur as a result of frequency changes. The proposed controller-based model's load current and load voltage THD values are 10.79% and 0.82%, respectively.

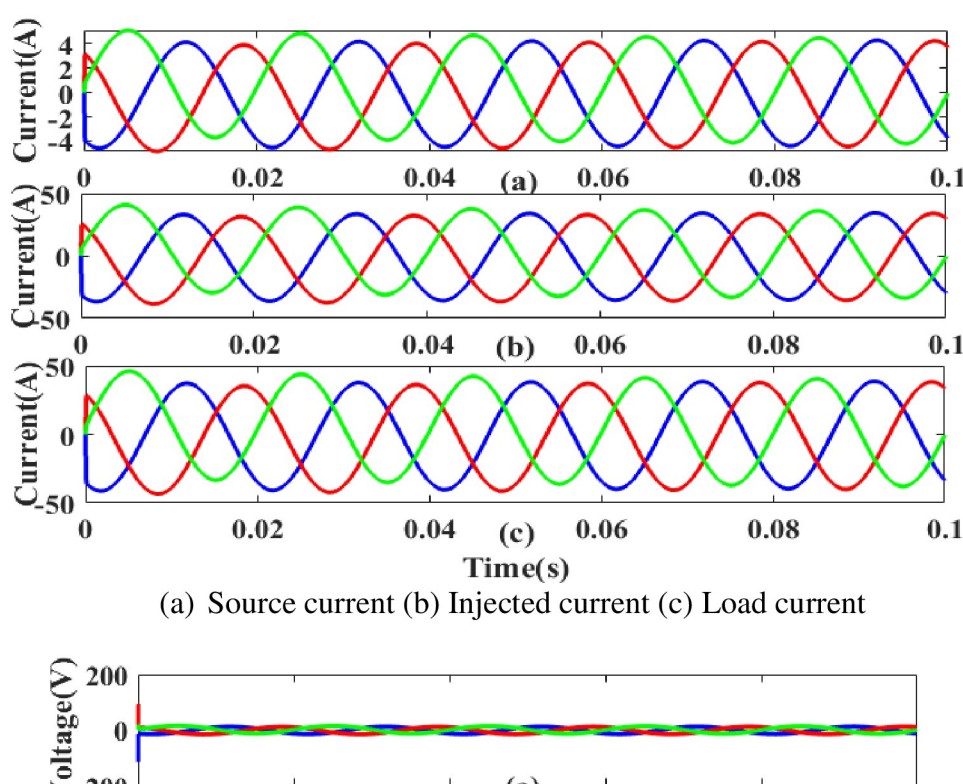

(a) Source current (b) Injected current (c) Load current

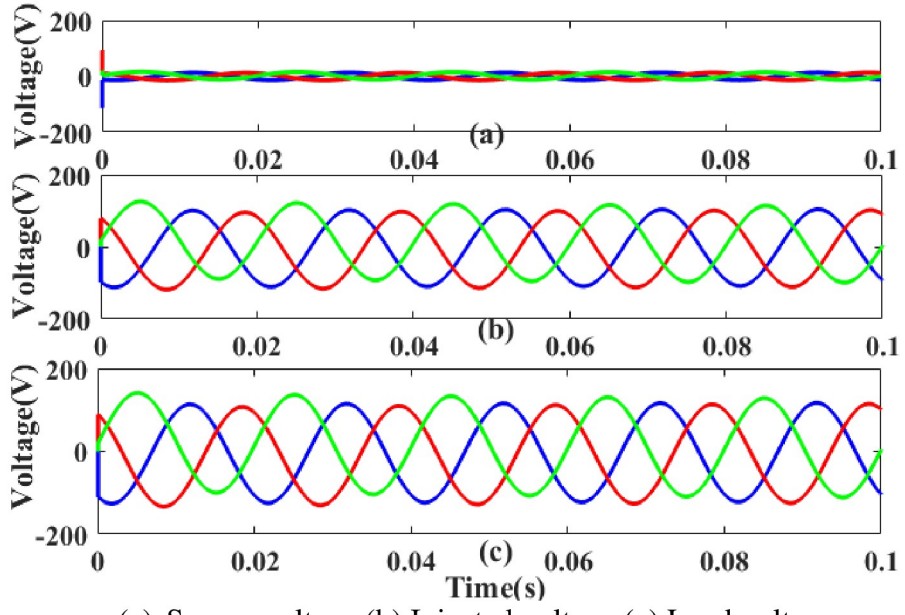

(a) Source voltage (b) Injected voltage (c) Load voltage

**Fig 19. Evaluation of current compensation and Voltage compensation at interruption period.** (a) Source current (b) Injected current (c) Load current. (a) Source voltage (b) Injected voltage (c) Load voltage.

**(b) Swell fault.** Swell is a poor power quality condition that adversely affects the load side when power is increased above the level of constant voltage. The proposed controller-based UPQC configuration's performance under scenario 3's swell conditions is shown in Fig 24. Current and voltage are injected during that period to balance out the load demand because the generated current has swell issues, as shown in Fig 24. In other words, although the supply current and voltage are flowing constantly and without interruption, there is a swell that arises each current and voltage is 0. 5 to 0.75 seconds. It is based on an intelligent controller.

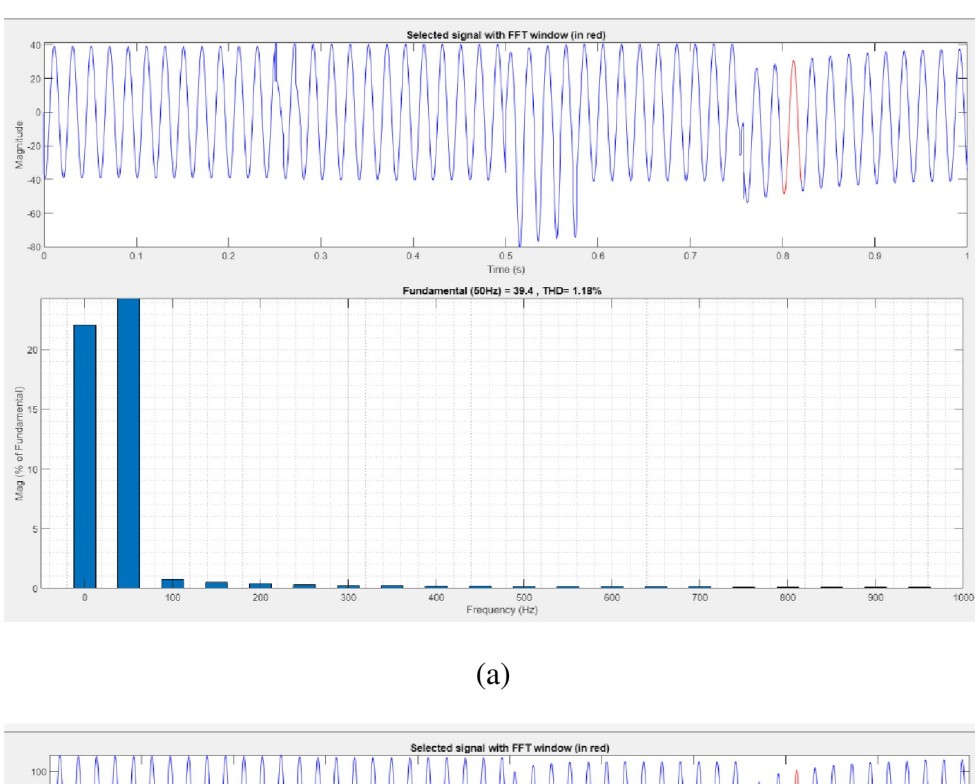

(a)

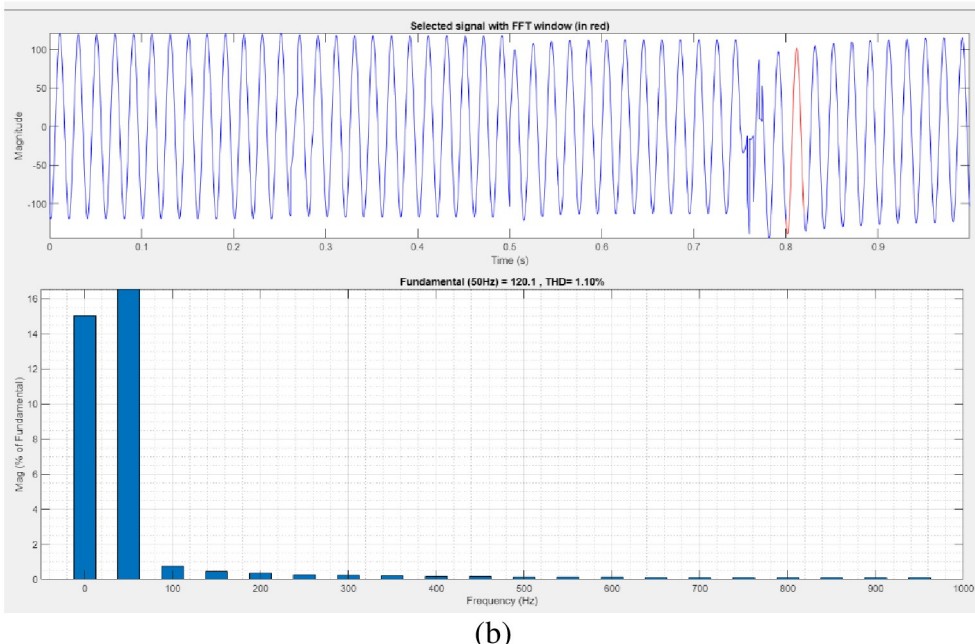

(b)

**Fig 20. THD analysis in (a) current and (b) voltage in interruption condition.**

Fig 25 illustrates an analysis of THD, harmonics in a system can occur as a result of frequency changes. The proposed controller-based model's load current and load voltage THD values are 10.18% and 0.48%, respectively.

**(c) Interruption fault.** Interruption is also regarded as one of the system's power quality issues its affects the components on the load side. Fig 26 depicts the interruption state, the injected current and voltage, and the rectified load current and load voltage. It demonstrates

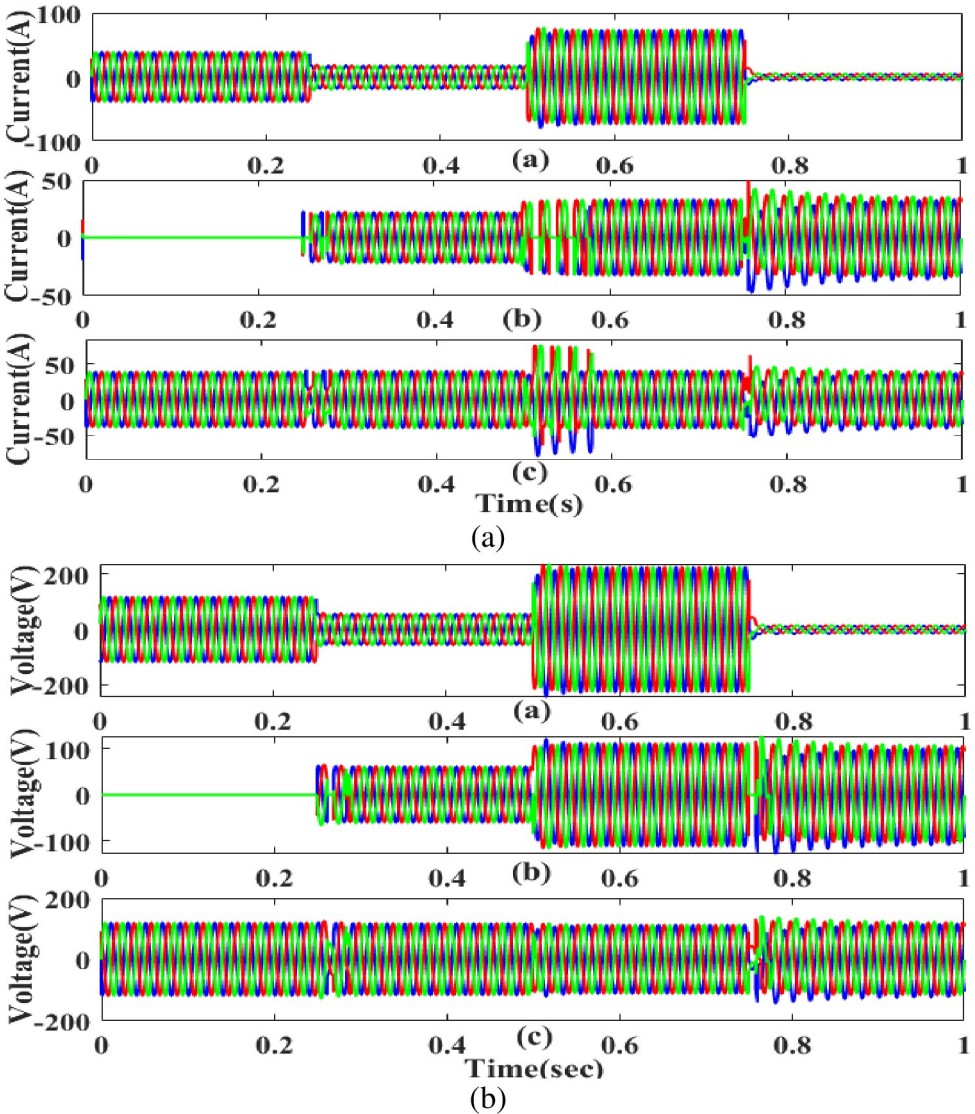

**Fig 21. Analysis of (a) 3-phase fault current compensation and (b) 3-phase fault voltage compensation.**

the interruption occurs to interrupt the power flow for 0.75 to 1 seconds based on the intelligent controller.

Fig 27 illustrates an analysis of THD, harmonics in a system can occur as a result of frequency changes. The load current and load voltage of THD of the proposed controller-based model is 10.07% and 1.01%, respectively.

**(d) Combined fault.** Fig 28 clearly shows the combined 3-phase current and voltage. 3-phase fault voltage and current interrupt the power flow in 0. 5 to 0.75 seconds based on the intelligent controller.

**Case 4: Static condition in nonlinear load**

In static conditions solar irradiance is considered as $800 w/m^2$ and temperature 25° integrated into the proposed optimal controller in nonlinear load is shown in Fig 29. Performance analysis is done in the proposed model is explained in the below section.

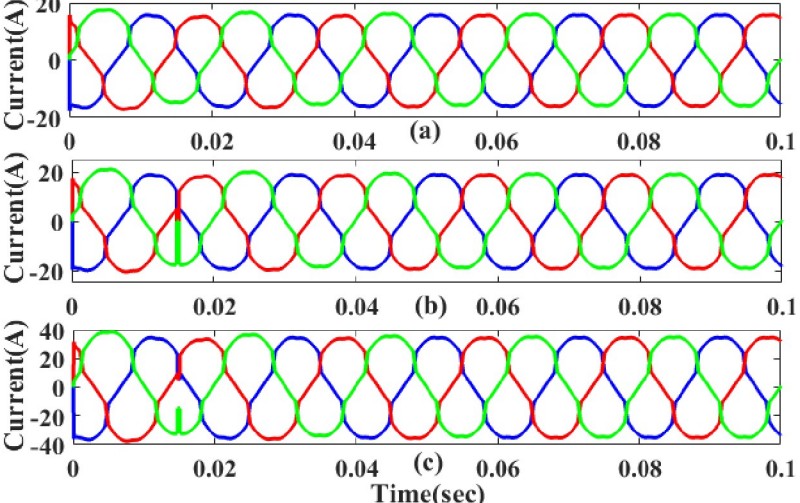

(a) Source current (b) Injected current (c) Load current

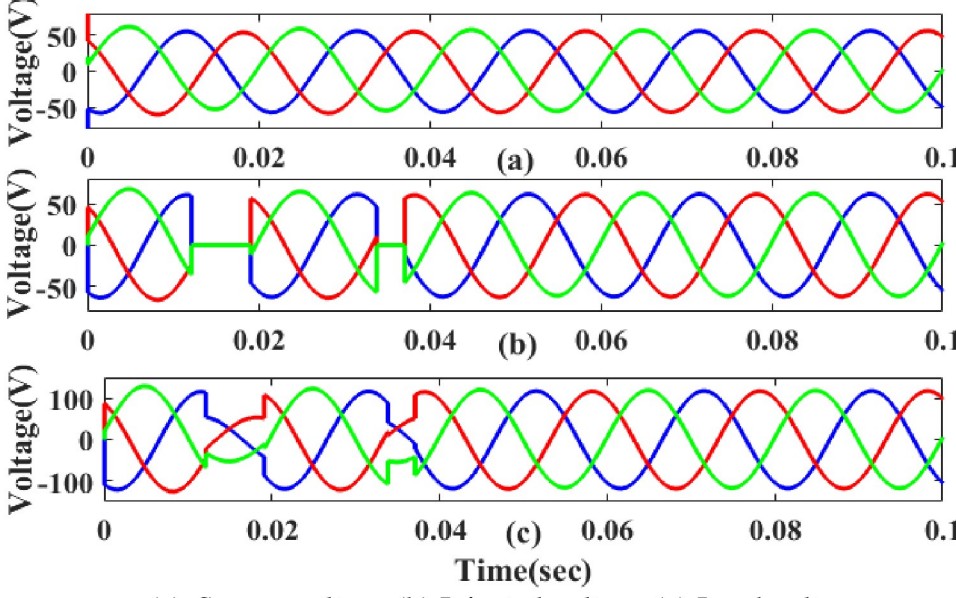

(a) Source voltage (b) Injected voltage (c) Load voltage

**Fig 22.** Evaluation of (a) current compensation and (b) Voltage compensation at Sag period. Source current (b) Injected current (c) Load current. Source voltage (b) Injected voltage (c) Load voltage.

**(a) Sag fault.** During the sag period in static condition in nonlinear load, the current and voltage decreases below its constant level, usually as a result of a systemic fault. Power was injected during that period, as shown in Fig 30. In other words, although the supply current and voltage are flowing constantly without interruption, there is a sag that arises each current and voltage is 0.25 to 0.5 seconds. It is based on an intelligent controller.

Fig 31 illustrates an analysis of THD, harmonics in a system can occur as a result of frequency changes. The load current and load voltage of THD of the proposed controller based model is 10.11% and 0.24%, respectively.

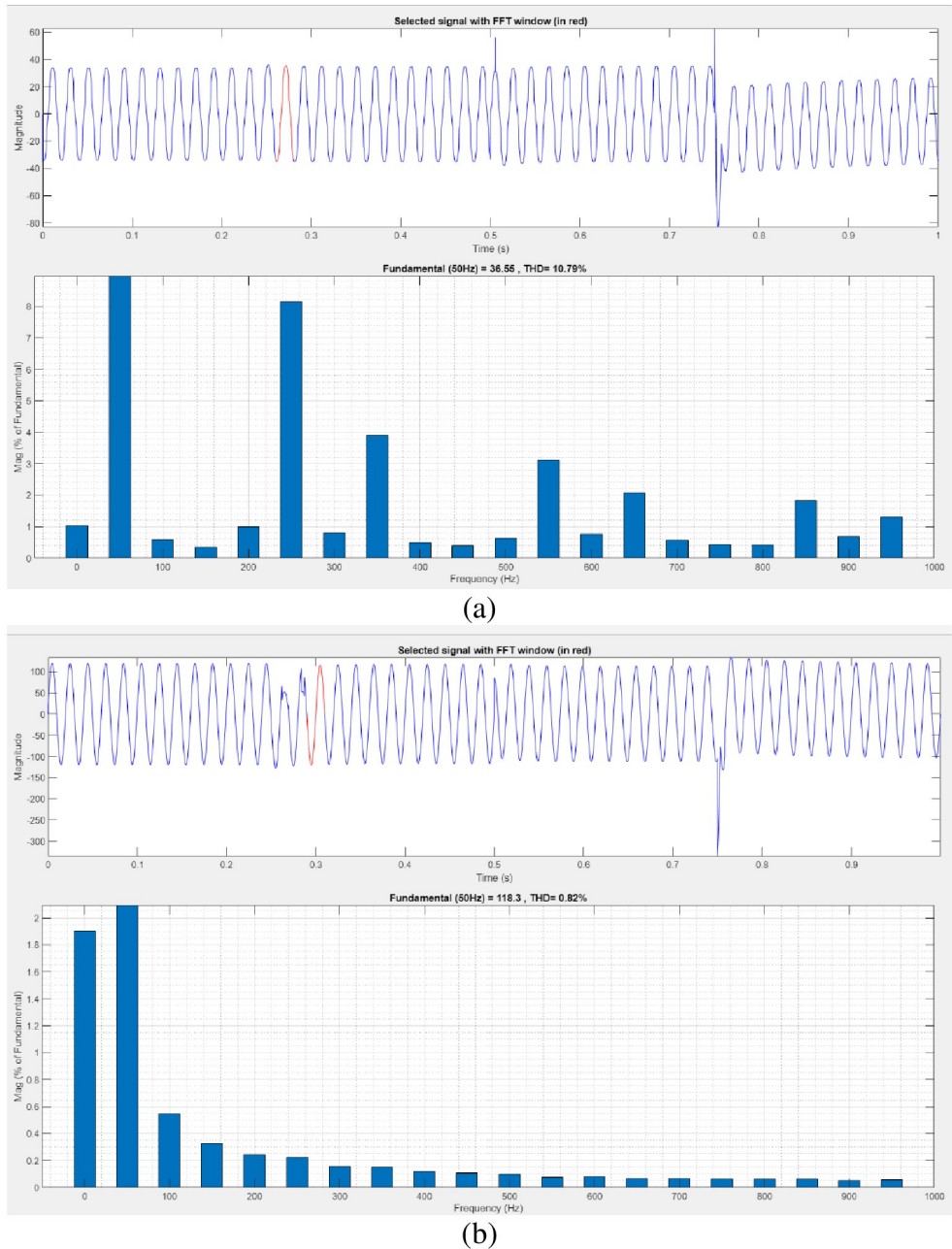

**Fig 23. THD analysis in (a) current and (b) voltage in sag condition.**

**(b) Swell fault.** Fig 32 displays the proposed controller based UPQC configuration under swell circumstances in case 2. Current and voltage are injected during that period to balance out the load demand because the generated current has swell issues, as shown in Fig 32. In other words, although the supply current and voltage are flowing constantly and without interruption, there is a swell that arises each current and voltage is 0. 5 to 0.75 seconds. It is based on an intelligent controller.

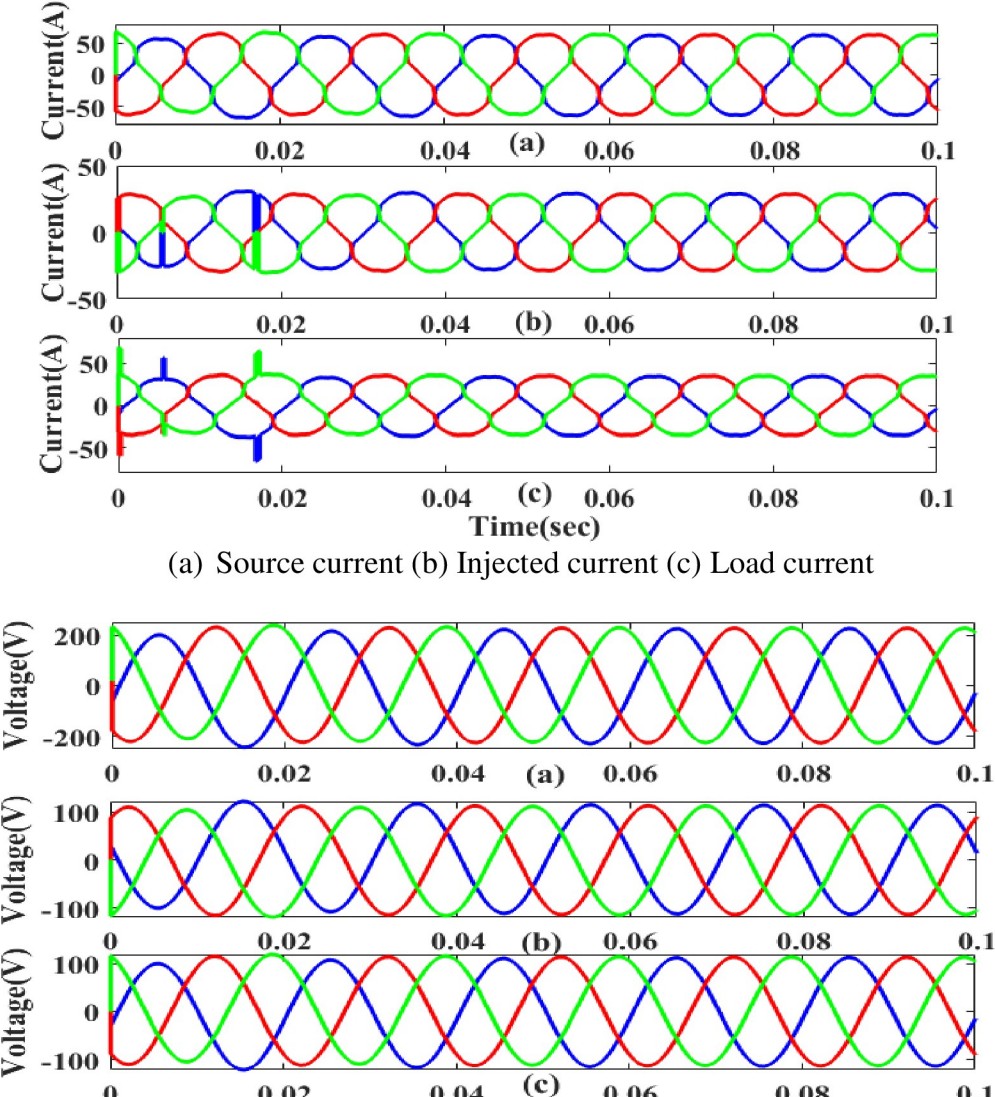

(a) Source current (b) Injected current (c) Load current

(a) Source voltage (b) Injected voltage (c) Load voltage

**Fig 24.** Evaluation of (a) current compensation and (b) Voltage compensation at swell period. (a) Source current (b) Injected current (c) Load current. (a) Source voltage (b) Injected voltage (c) Load voltage.

Fig 33 illustrates an analysis of total harmonic distortion (THD). Harmonics in a system can occur as a result of frequency changes. The load current and load voltage of THD of the proposed controller based model is 10.08% and 0.44%, respectively.

**(c) Interruption fault.** Fig 34 depicts the interruption state, the injected current and voltage, as well as the load current and load voltage that have been compensated. It demonstrates the interruption occurs to interrupt the power flow for 0.75 to 1 seconds based on the intelligent controller.

Fig 35 illustrates an analysis of total harmonic distortion (THD). Harmonics in a system can occur as a result of frequency changes. The load current and load voltage of THD of the proposed controller-based model is 10.07% and 0.62%, respectively.

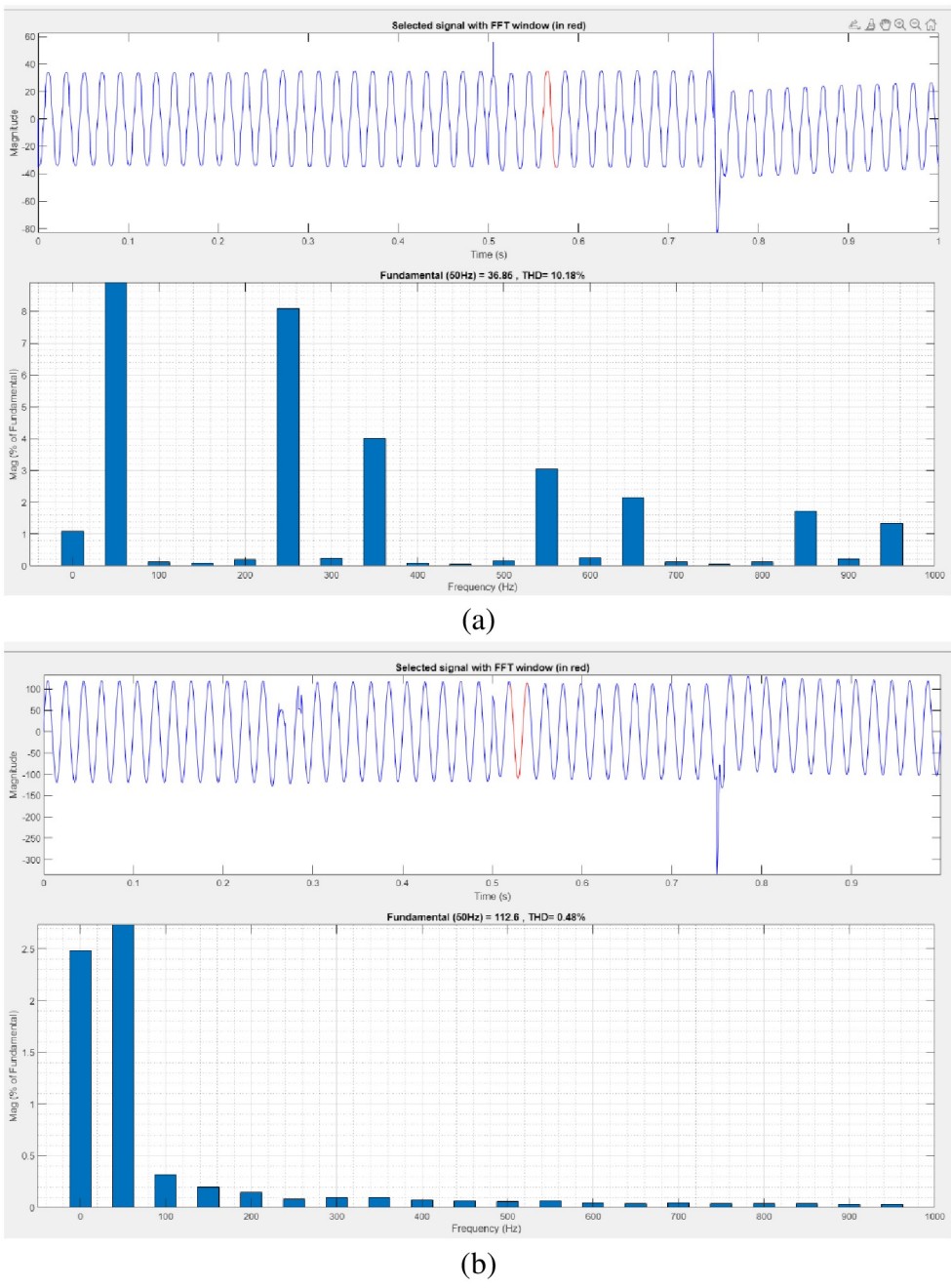

**Fig 25. THD analysis in (a) current and (b) voltage in swell condition.**

**(d) Combined fault.** Fig 36 clearly shows the combined 3-phase current and voltage.3phase fault voltage and current interrupt the power flow in 0. 5 to 0.75 seconds based on the intelligent controller.

**Case 5**: **Variation in load parameters**

In the case 5 condition, solar irradiance is taken as $1000w/m^2$ and temperature 25˚ integrated in the proposed optimal controller in nonlinear load and parameter variation is considered. Performance analysis is done in the proposed model is explained in the below section.

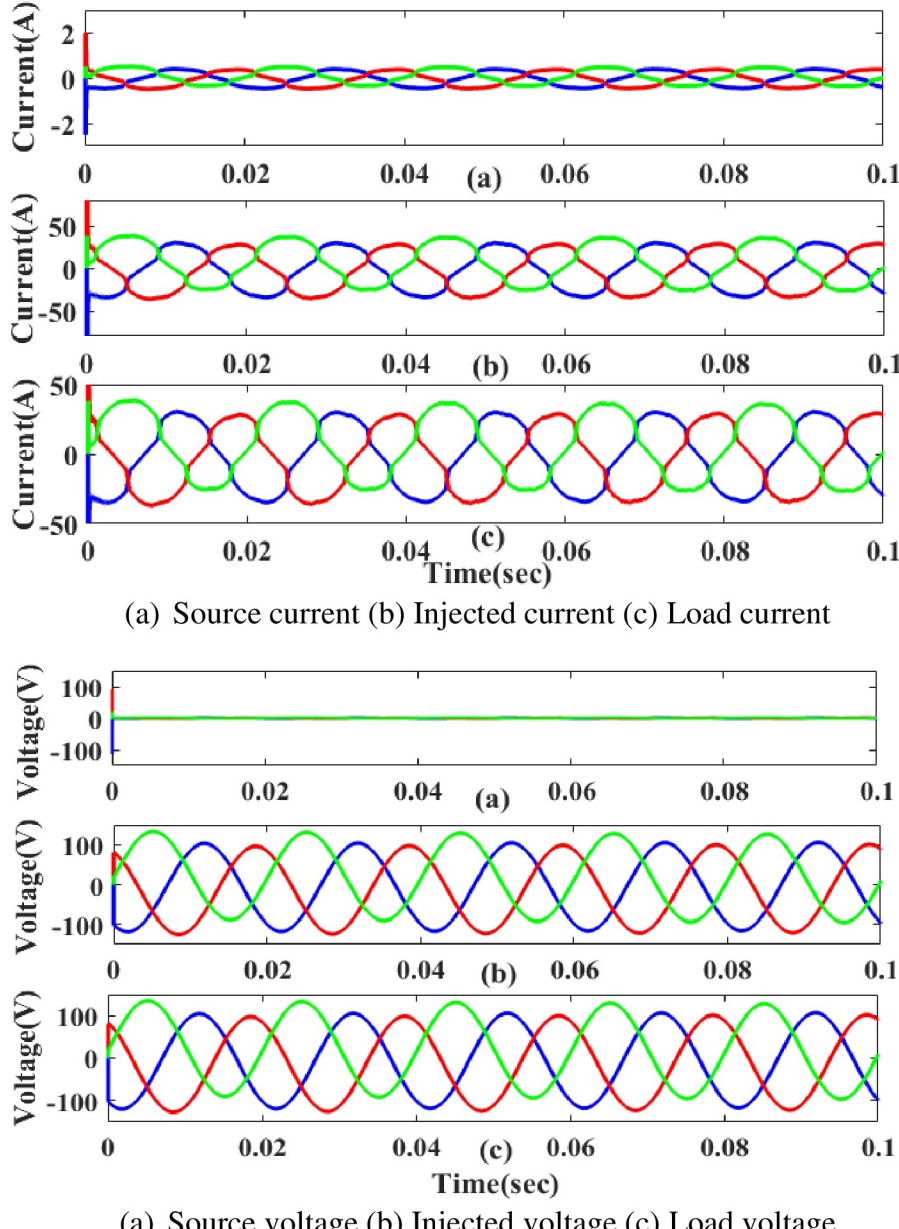

(a) Source current (b) Injected current (c) Load current

(a) Source voltage (b) Injected voltage (c) Load voltage

**Fig 26.** Evaluation of (a) current compensation and (b) Voltage compensation at interruption period. (a) Source current (b) Injected current (c) Load current. (a) Source voltage (b) Injected voltage (c) Load voltage.

**(a) Sag fault.** Due to generated current sag problems, as depicted in Fig 37, power is injected during that time to balance off the load demand. In other words, although the supply current and voltage are flowing constantly without interruption, there is a sag that arises each current and voltage is 0.25 to 0.5 seconds. It is based on an intelligent controller.

Fig 38 illustrates an analysis of THD harmonics in a system that can occur as a result of frequency changes. The load current and load voltage of THD of the proposed controller based model is 10.23% and 0.27%, respectively.

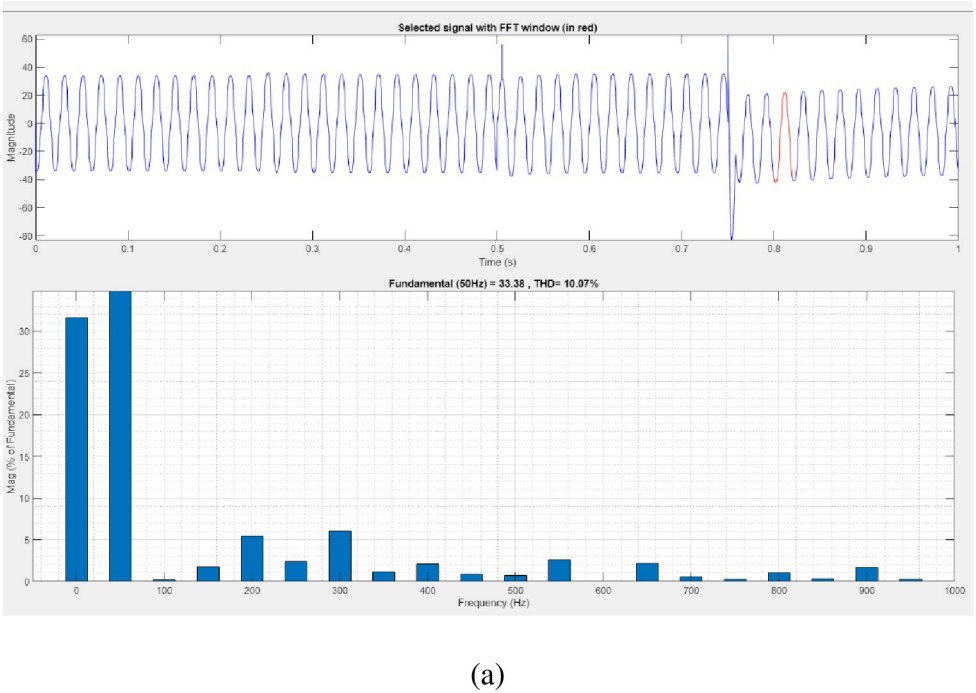

(a)

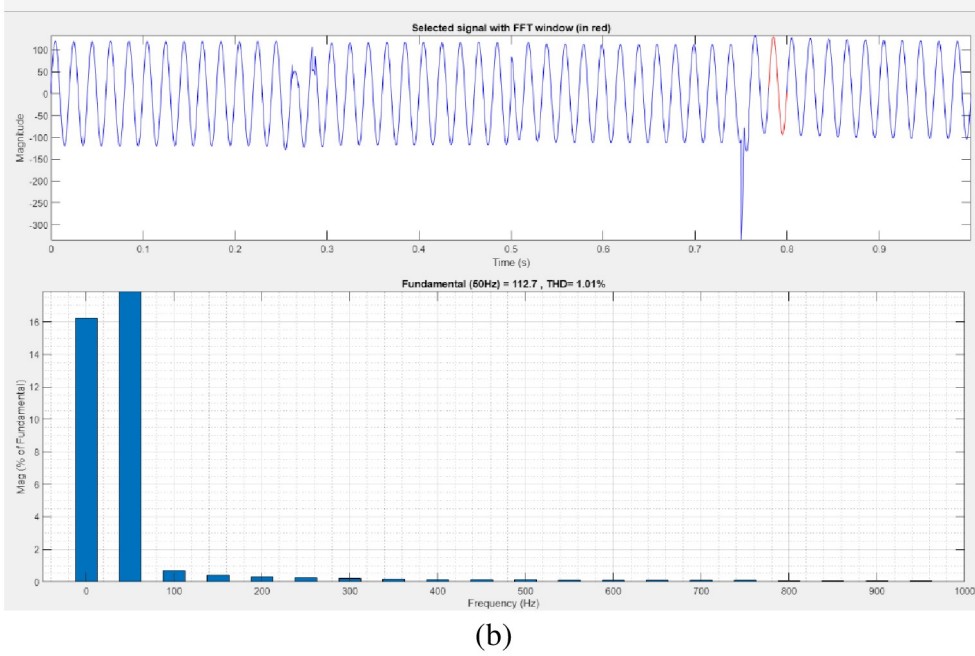

(b)

**Fig 27. THD analysis in (a) current and (b) voltage in interruption condition.**

**(b) Swell fault.** Current and voltage are injected during that period to balance out the load demand because the generated current has swell issues, as shown in Fig 39. In other words, although the supply current and voltage are flowing constantly and without interruption, there is a swell that arises each current and voltage is 0. 5 to 0.75 seconds. It is based on an intelligent controller.

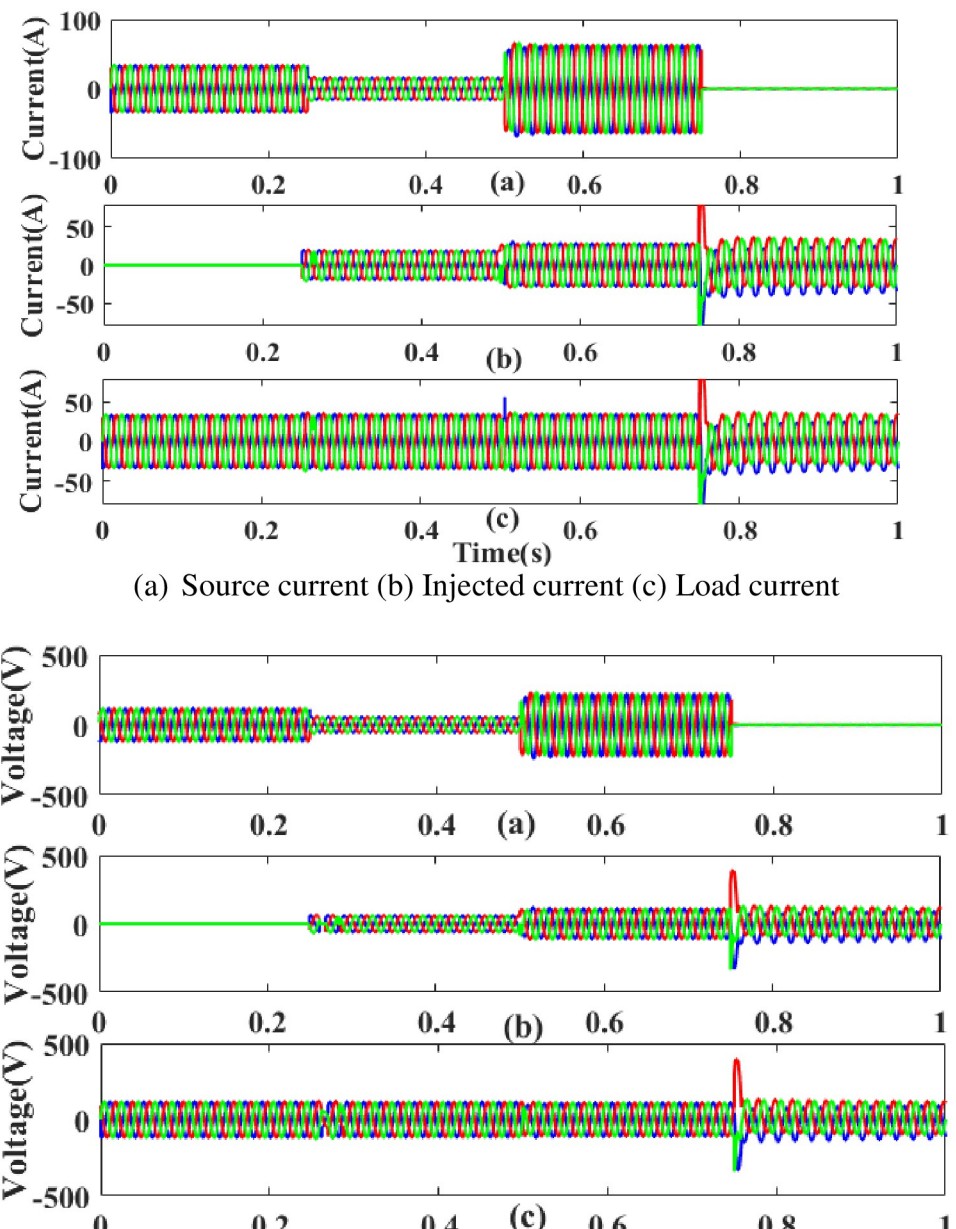

(a) Source current (b) Injected current (c) Load current

(a) Source voltage (b) Injected voltage (c) Load voltage

**Fig 28. Analysis of 3phase fault current compensation and 3phase fault voltage compensation.** (a) Source current (b) Injected current (c) Load current. (a) Source voltage (b) Injected voltage (c) Load voltage.

Fig 40 illustrates an analysis of total harmonic distortion (THD). Harmonics in a system can occur as a result of frequency changes. The load current and load voltage of THD of the proposed controller based model is 10.23% and 0.26%, respectively.

**(c) Interruption fault.** The interruption condition, the injected current and voltage are compensated load current and load voltage are shown in Fig 41. It demonstrates the interruption occurs to interrupt the power flow for 0.75 to 1 seconds based on the intelligent controller.

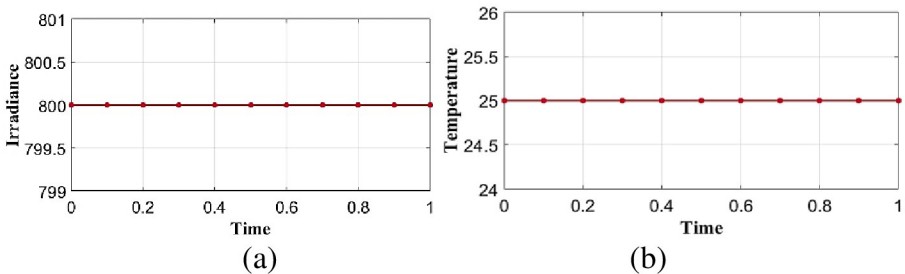

**Fig 29. Analysis of (a) solar irradiance and (b) temperature with respect to time in case 4.**

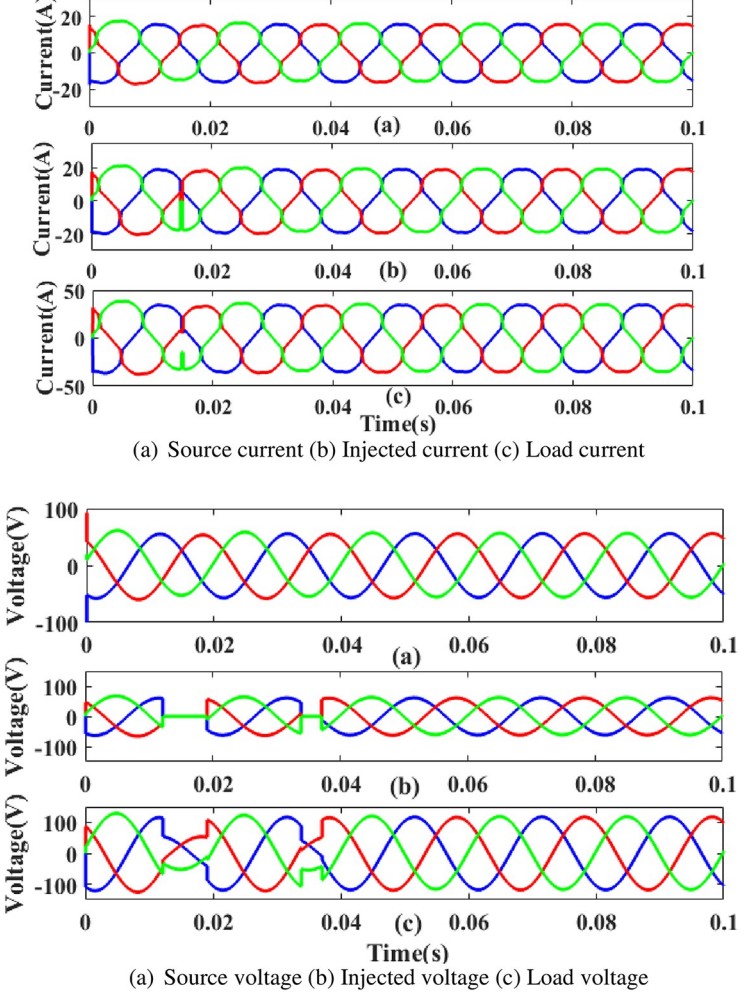

(a) Source current (b) Injected current (c) Load current

(a) Source voltage (b) Injected voltage (c) Load voltage

**Fig 30.** Evaluation of (a) current compensation and (b) Voltage compensation at sag period. (a) Source current (b) Injected current (c) Load current. (a) Source voltage (b) Injected voltage (c) Load voltage.

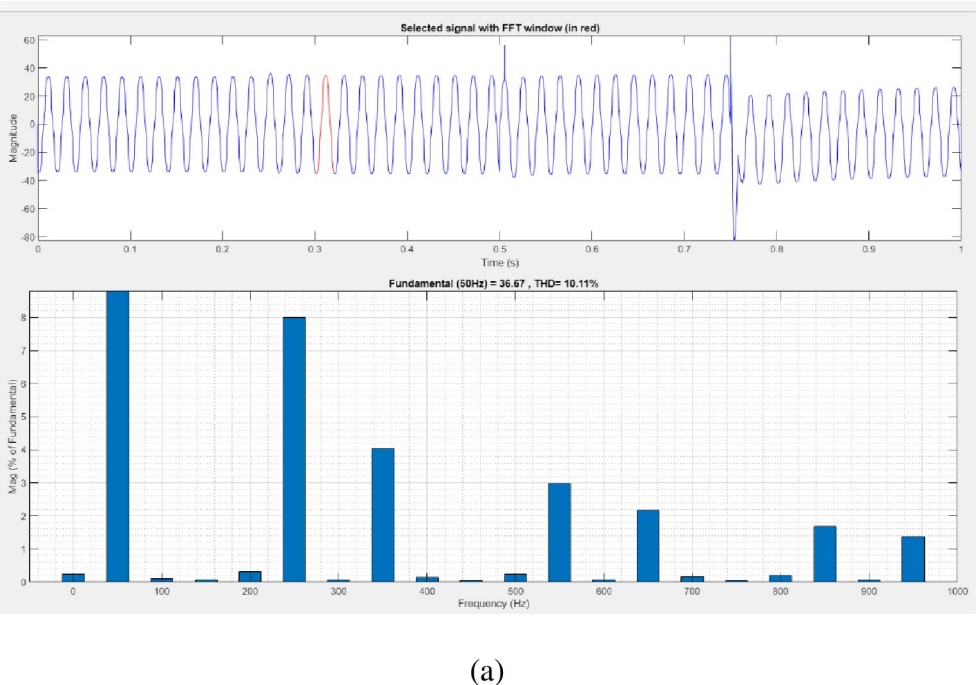

(a)

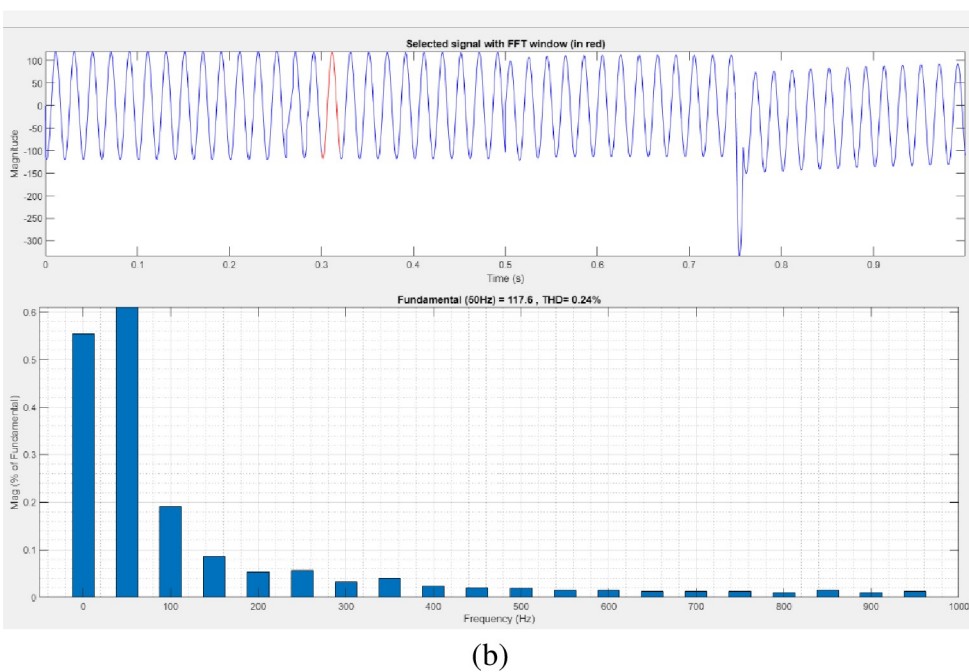

(b)

**Fig 31. THD analysis in (a) current and (b) voltage in sag condition.**

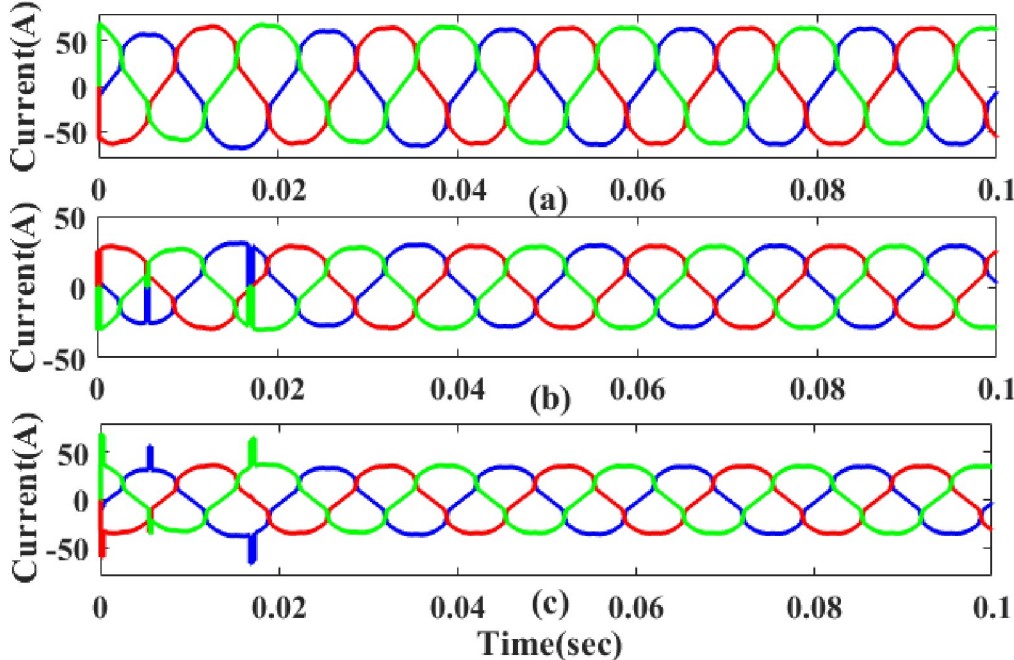

(a) Source current (b) Injected current (c) Load current

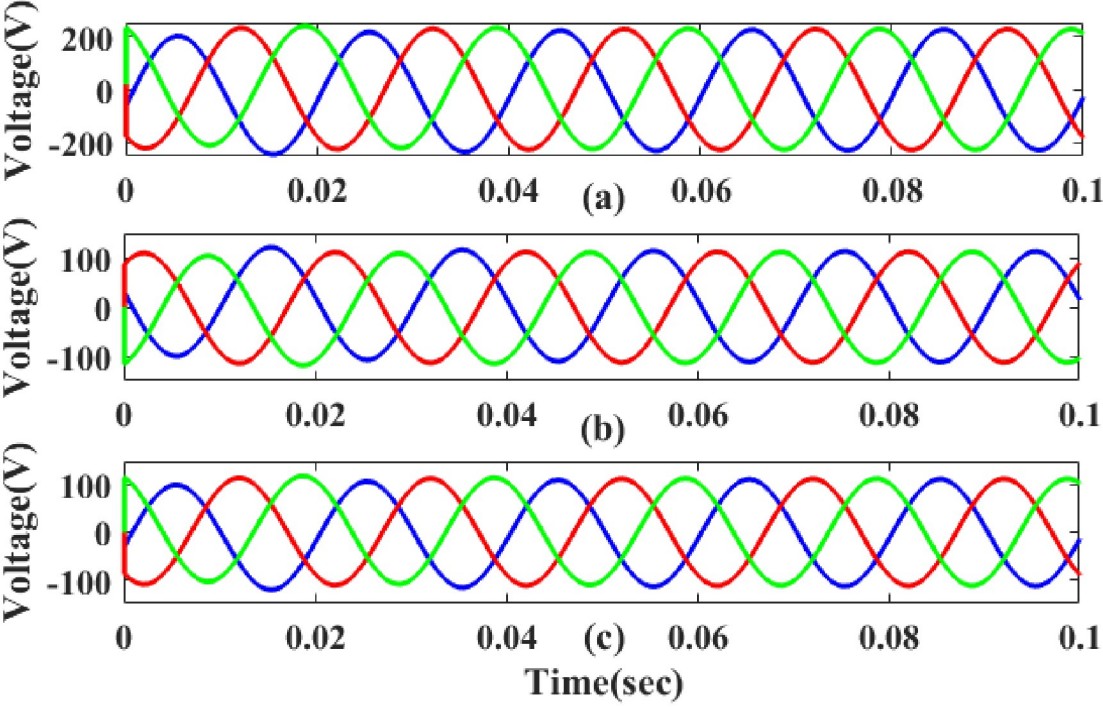

(a) Source voltage (b) Injected voltage (c) Load voltage

**Fig 32. Evaluation of current compensation and voltage compensation at swell period.** (a) Source current (b) Injected current (c) Load current. (a) Source voltage (b) Injected voltage (c) Load voltage.

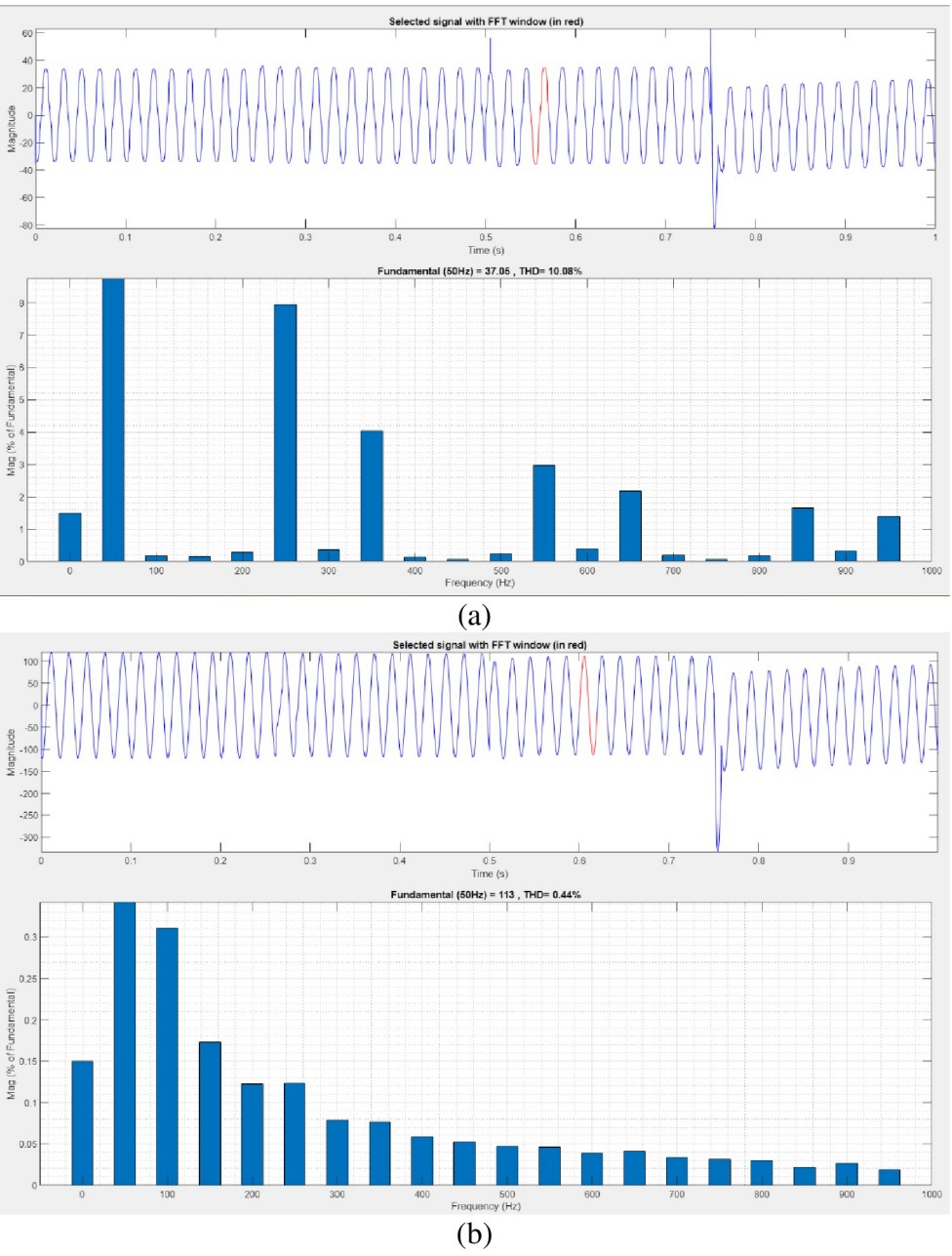

**Fig 33. THD analysis in (a) current and (b) voltage in swell condition.**

Fig 42 illustrates an analysis of THD harmonics in a system that can occur as a result of frequency changes. The load current and load voltage of THD of the proposed controller based model is 10.34% and 0.36%, respectively.

**(d) Combined fault.** Fig 43 clearly shows the combined 3-phase current and voltage.3phase fault voltage and current interrupt the power flow in 0. 5 to 0.75 seconds based on the intelligent controller.

Fig 44 clearly shows the comparison of convergence in proposed and existing algorithms like the Aquila Optimization Algorithm (AOA), Pelican Optimization Algorithm (POA),

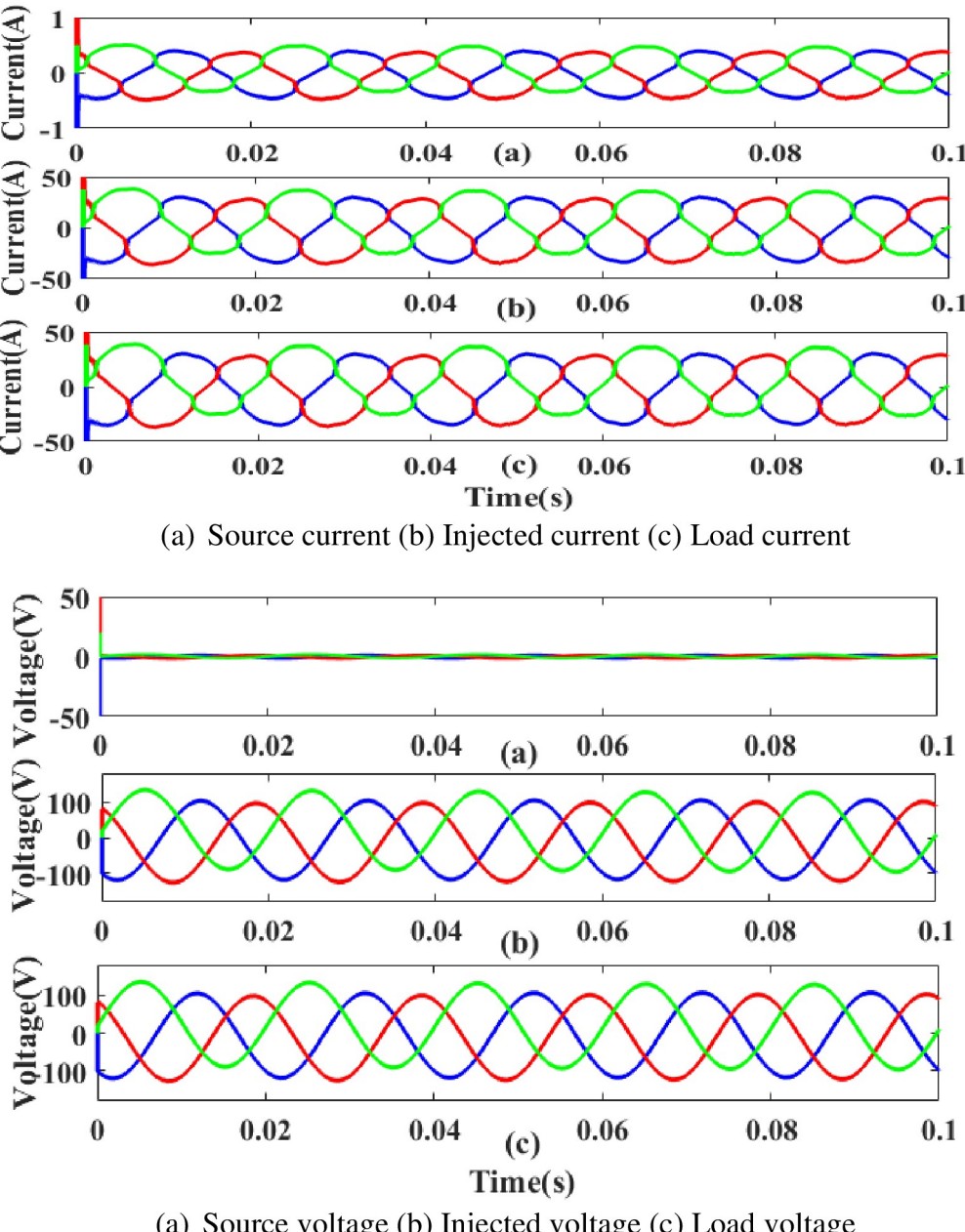

(a) Source current (b) Injected current (c) Load current

(a) Source voltage (b) Injected voltage (c) Load voltage

**Fig 34. Evaluation of (a) current compensation and (b) Voltage compensation at interruption period.** (a) Source current (b) Injected current (c) Load current. (a) Source voltage (b) Injected voltage (c) Load voltage.

African Vulture Optimization (AVO) and Ant Lion Optimization (ALO) Algorithm. If the relevant sequence converges for specified initial approximations, an iterative process is said to be convergent.

In the proposed hybrid algorithm, the 18th iteration to overcome the error is 130. In existing methods, AOA attains the 10th error state in the 7th iteration, POA attains the 40th error state in the 5th iteration, AVO attains the 35th error state in the 10th iteration and ALO attains

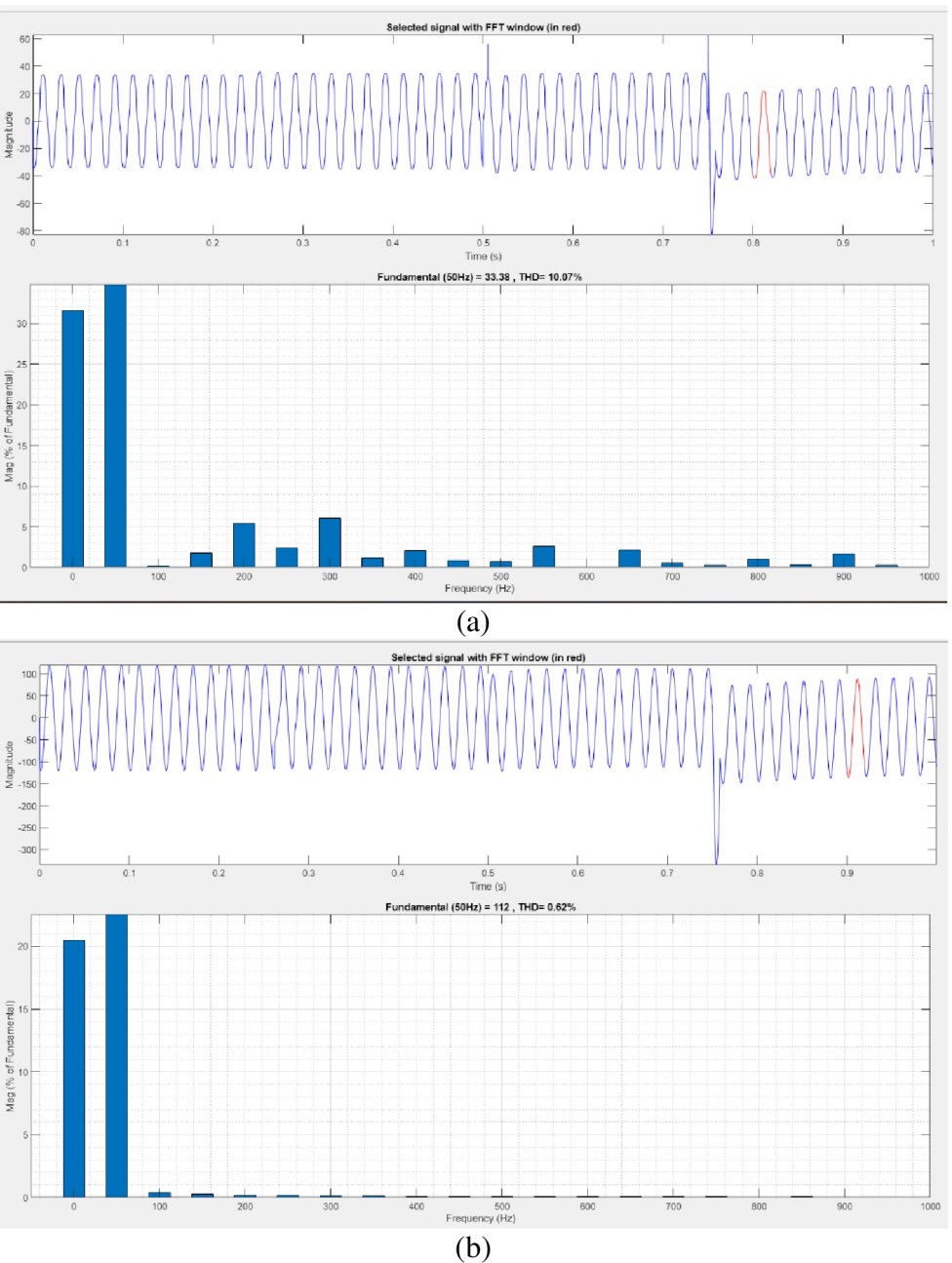

**Fig 35. THD analysis in (a) current and (b) voltage in interruption condition.**

the 9th error in the 15th iteration. Compared to existing methods, the proposed method effectively performs.

**DC–link voltage.** While active/reactive power control offers direct control of power injection into the grid, DC-link voltage control modulates dc-link voltages to control power. To demonstrate the way the controllers in Fig 45 are performing, simulation results are given. During fluctuations in load and irradiation, the proposed controller keeps the stable voltage across the DC connection constant at 690V.

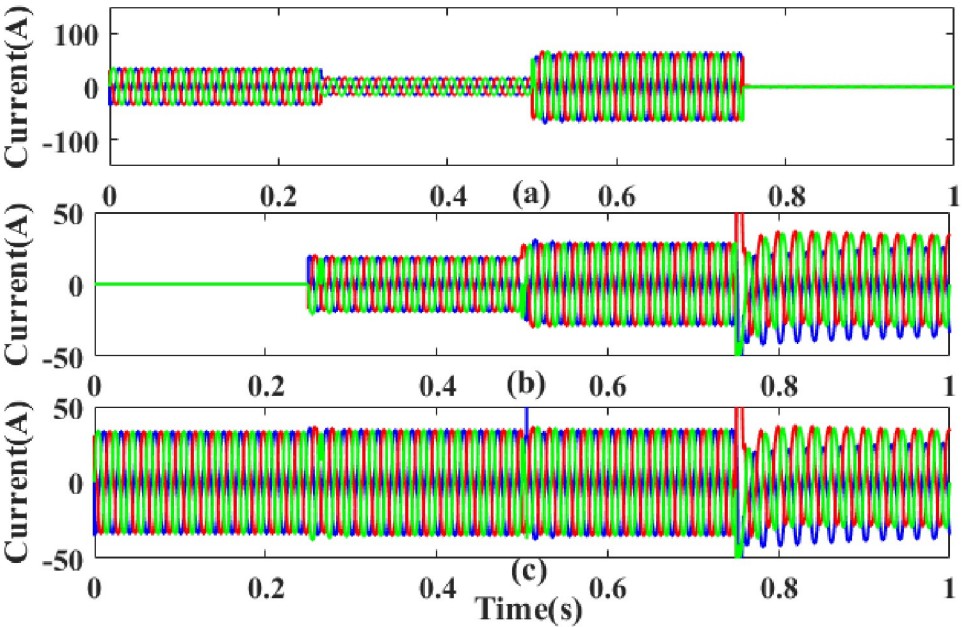

(a) Source current (b) Injected current (c) Load current

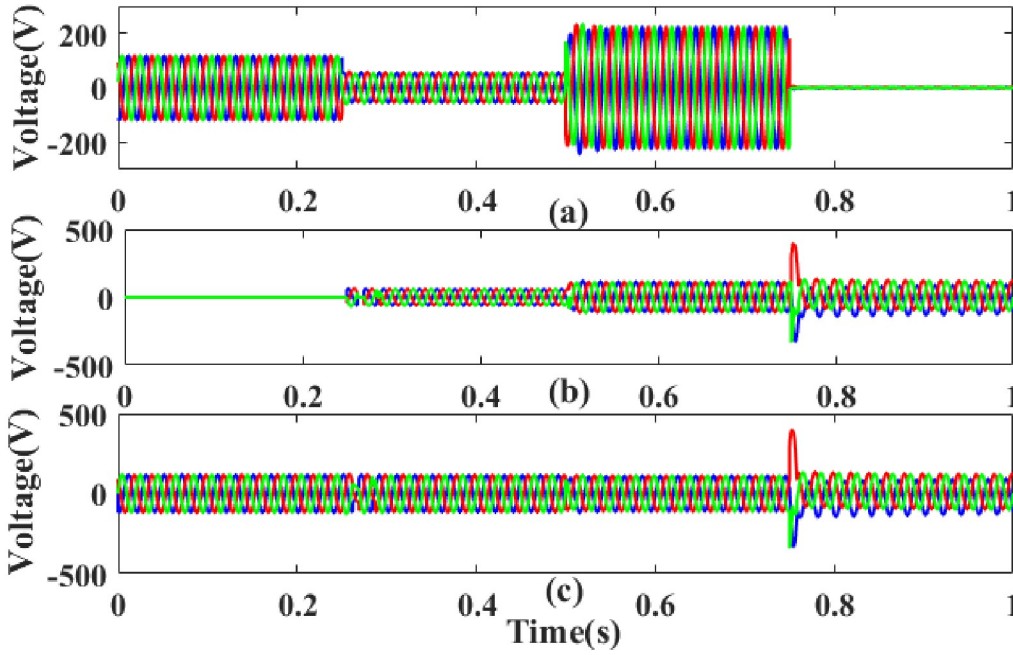

(a) Source voltage (b) Injected voltage (c) Load voltage

**Fig 36.** Analysis of (a) 3phase fault current compensation (b) 3-phase fault voltage compensation at interruption period. (a) Source current (b) Injected current (c) Load current. (a) Source voltage (b) Injected voltage (c) Load voltage.

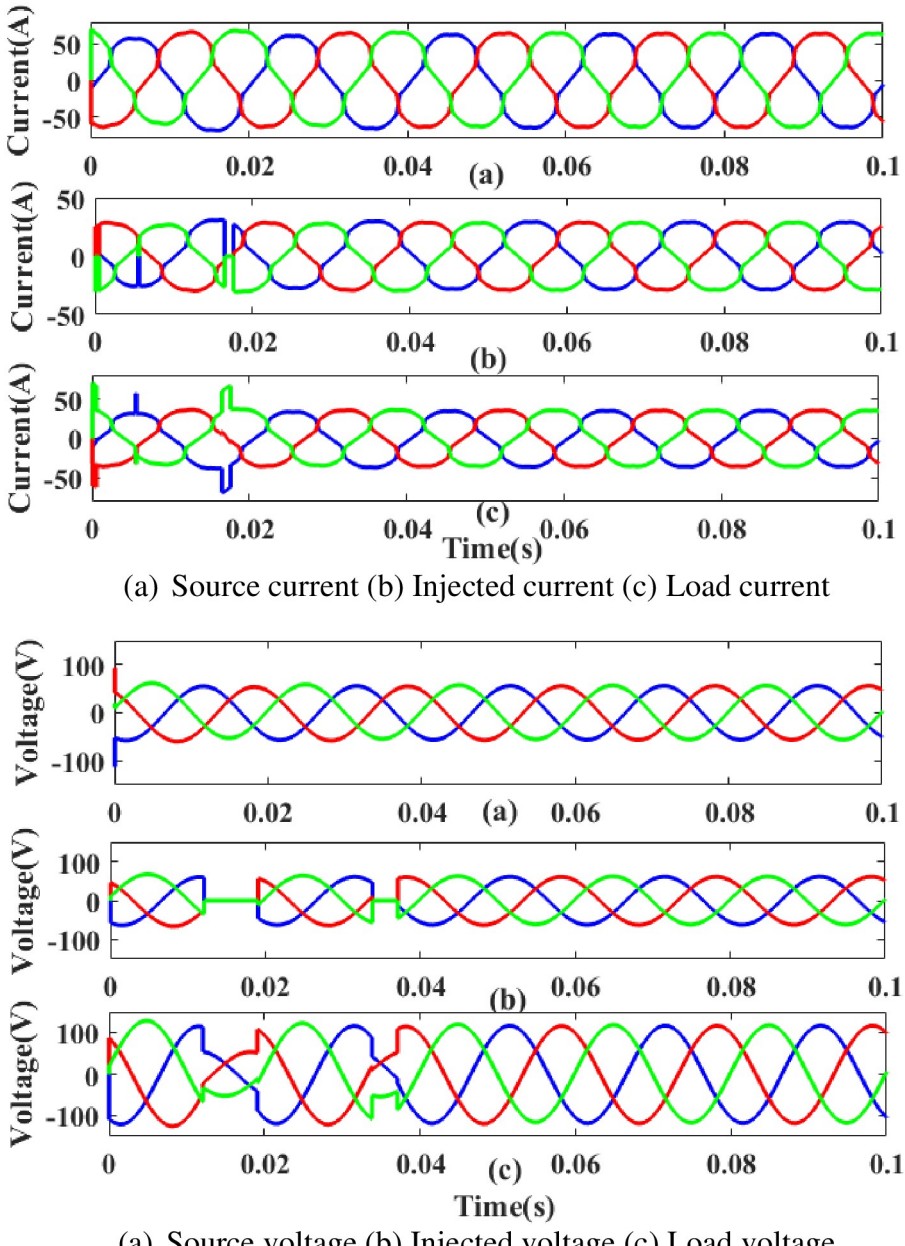

(a) Source current (b) Injected current (c) Load current

(a) Source voltage (b) Injected voltage (c) Load voltage

**Fig 37. Evaluation of current compensation and voltage compensation at sag period.** (a) Source current (b) Injected current (c) Load current. (a) Source voltage (b) Injected voltage (c) Load voltage.

### Comparison of performance in the proposed controller

In the following part, clearly explain how the proposed and existing controllers PF, test cases, and THD in specific scenarios compare in Tables 4–6. The proposed model effectively works compared to the current approaches.

In below Table 5 clearly shows the comparison of Power Factor (PF) in proposed and existing methods in individual cases. Compared to existing methods, the proposed controller PF range is 0.9 in all cases.

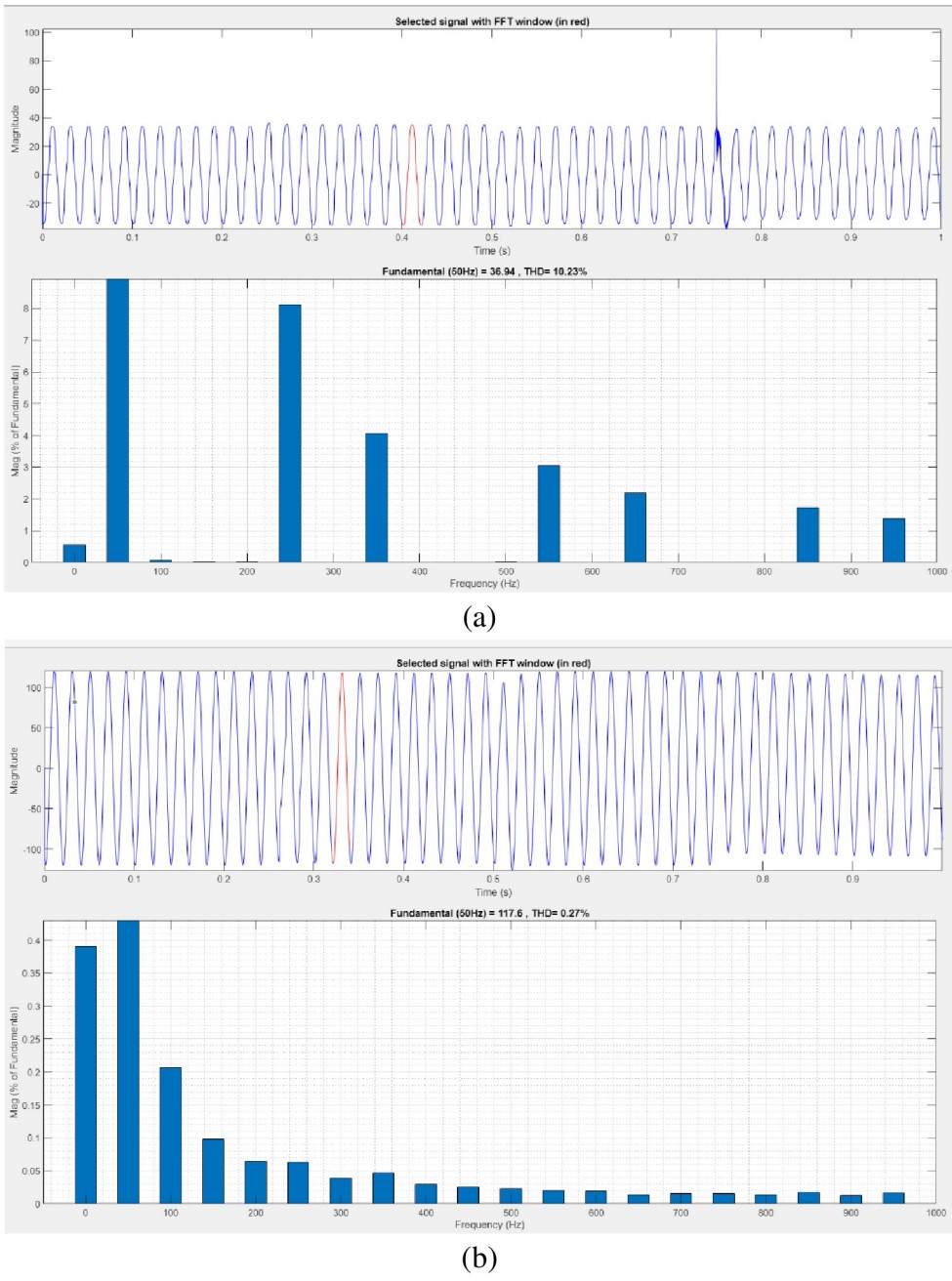

**Fig 38. THD analysis in (a) current and (b) voltage in sag condition.**

As indicated in Table 6, five test investigations using various combinations of linear and nonlinear loads, variable irradiation, supply voltages, and circumstances like sag, disturbance and swell were taken into consideration to attest to the performance of the suggested controller.

Total harmonic distortion (THD), a measurement of the harmonic distortion contained in a signal, is defined as the ratio of the sum of powers of all harmonic components to the power

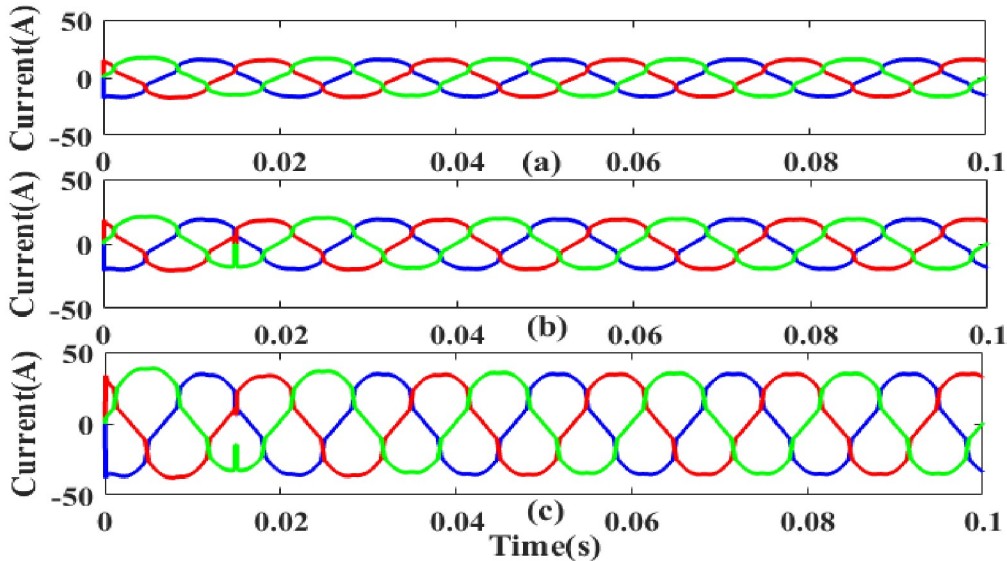

(a) Source current (b) Injected current (c) Load current

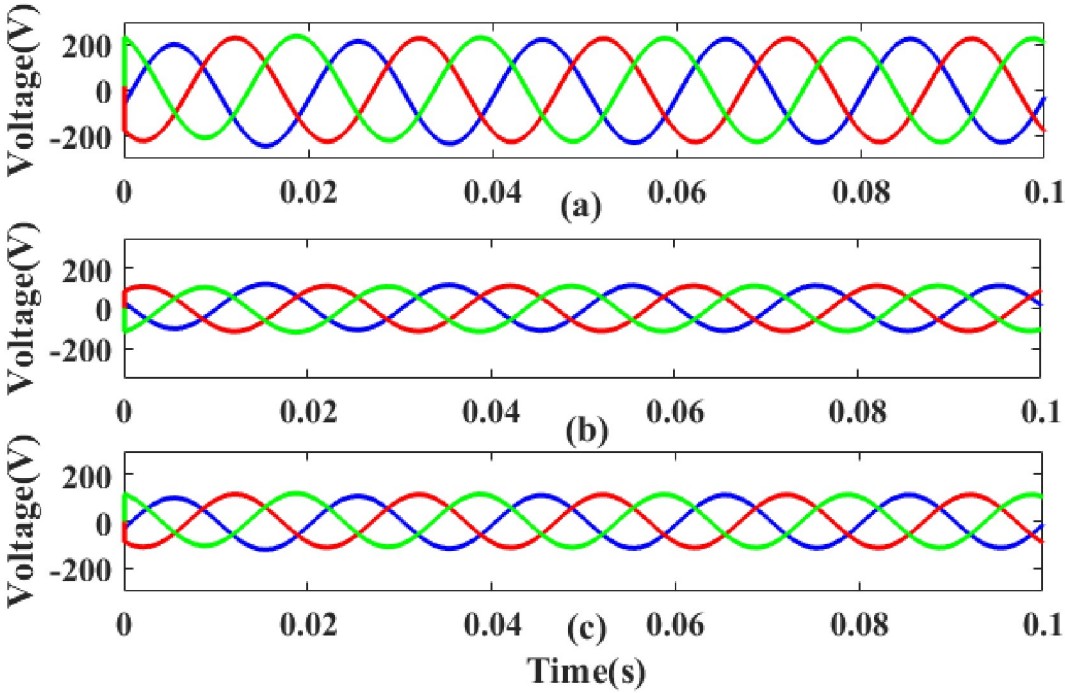

(a) Source voltage (b) Injected voltage (c) Load voltage

**Fig 39. Evaluation of current compensation and voltage compensation at swell period.** (a) Source current (b) Injected current (c) Load current. (a) Source voltage (b) Injected voltage (c) Load voltage.

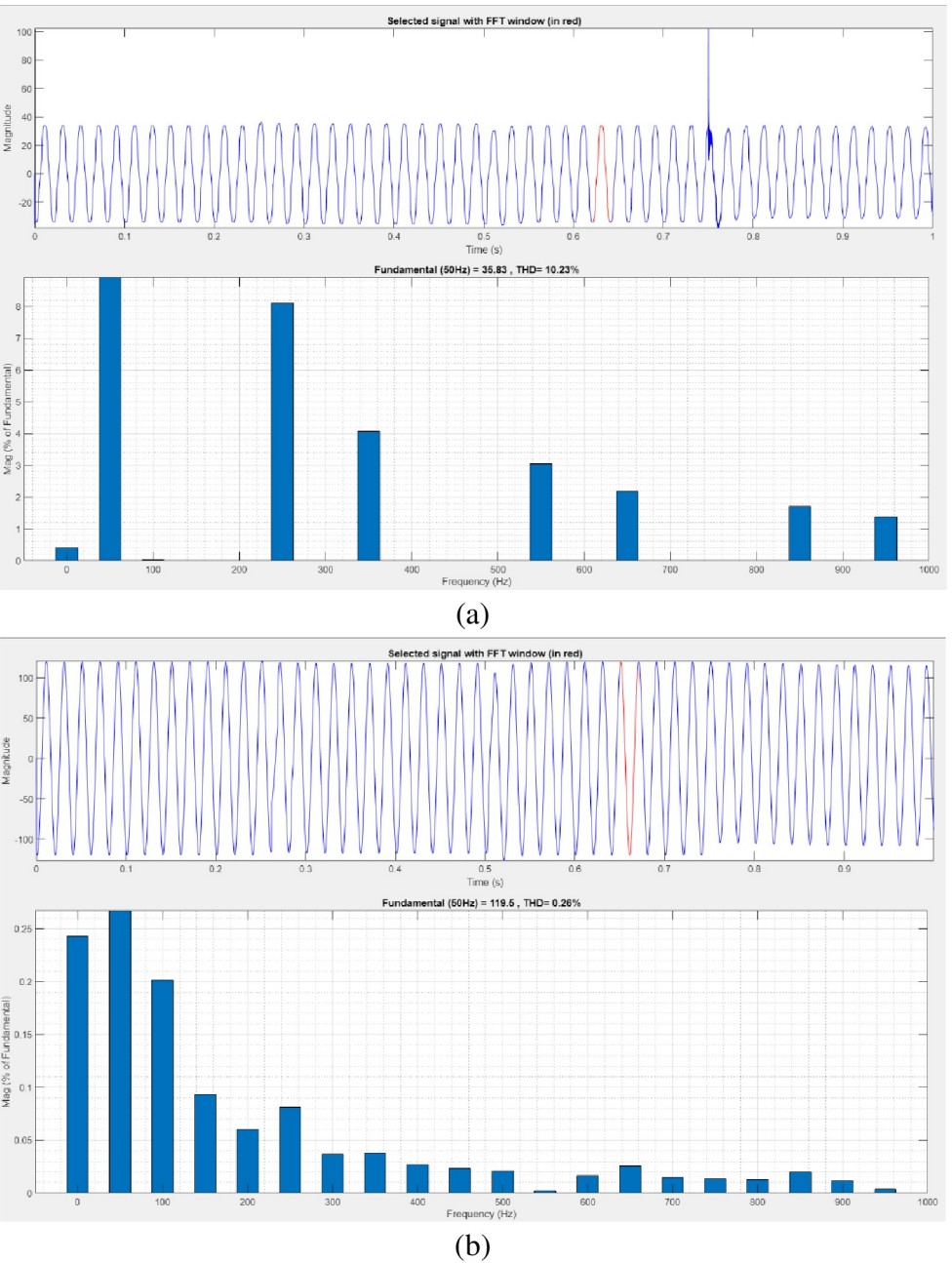

**Fig 40. THD analysis in (a) current and (b) voltage in swell condition.**

of the fundamental frequency. The above Table 7 clearly shows the comparison of THD in the proposed and existing controller.

## 5. Conclusion

An advanced optimum controller based on UPQC to reduce harmonics, sag, swell, and interruption in RES was proposed. PQ problems are regarded as the primary issue in the power system. Low PQ from the source, which has an impact on the power of electronic devices and

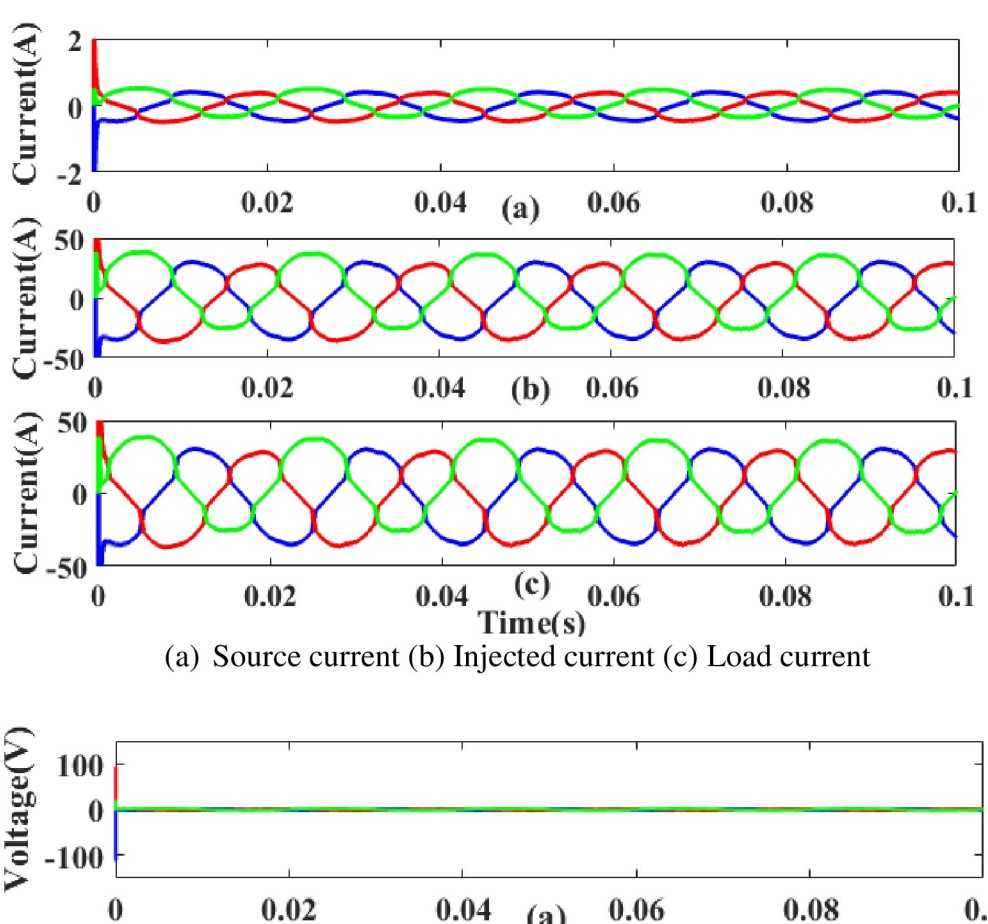

(a) Source current (b) Injected current (c) Load current

(a) Source voltage (b) Injected voltage (c) Load voltage

**Fig 41.** Evaluation of (a) current (b) Voltage compensation at interruption period. (a) Source current (b) Injected current (c) Load current. (a) Source voltage (b) Injected voltage (c) Load voltage.

loads, causes the voltage sag and swell on the consumer side. PQ enhancement in RES grid-connected systems has advanced as a study related to DG integrated with RES systems in order to address PQ concerns. Non-linear loads, unpredictable loads, and high-frequency switching characteristics make PQ issues in the device worse. So, in the proposed work, a hybrid

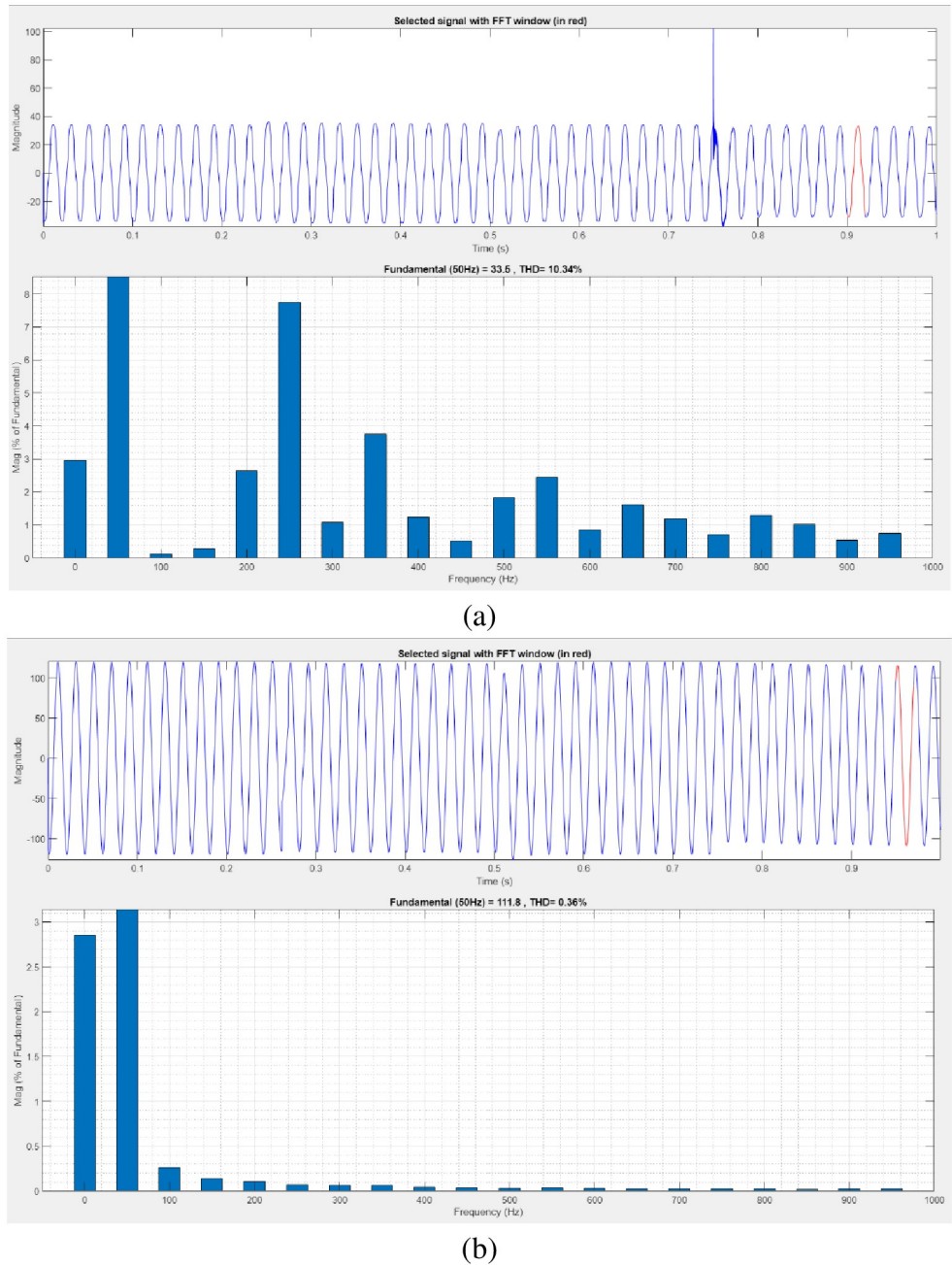

(a)

(b)

**Fig 42. THD analysis in (a) current and (b) voltage in interruption condition.**

optimization based FOPID controller was developed to mitigate PQ issues under various conditions. In the proposed model RES system, which has a three-phase load on the distribution side PV, batteries, and the grid on the source side. To control the power flows, a connection is formed between the load and supply sides of a UPQC, which controls both current and voltage variations. The control of UPQC is done through the use of the FOPID controller. FOPID controller is developed to generate command signals by analyzing the variation in load bus power in order to enhance UPQC's mitigation performance. FOPID have five various parameters

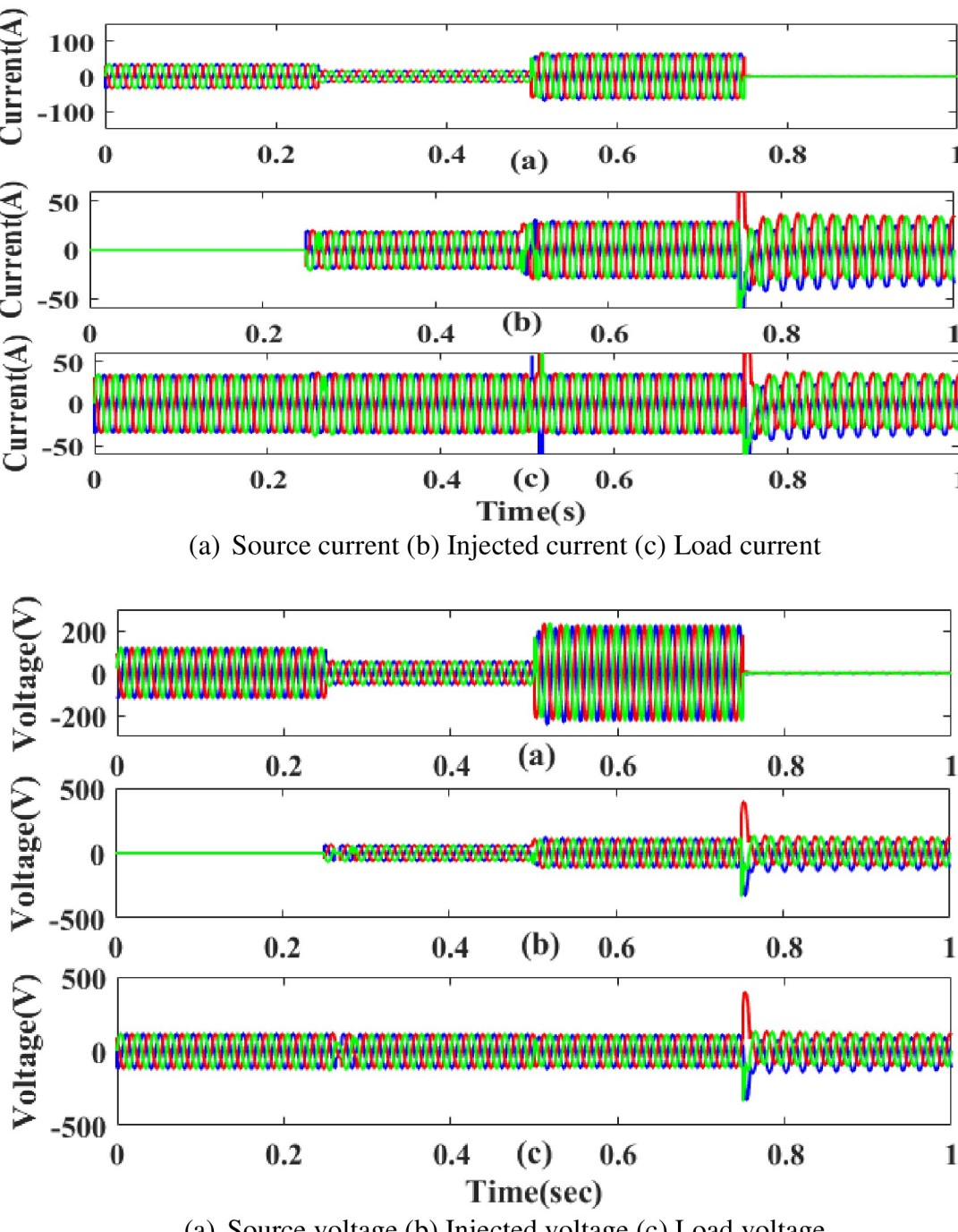

(a) Source current (b) Injected current (c) Load current

(a) Source voltage (b) Injected voltage (c) Load voltage

**Fig 43.** Evaluation of (a) current (b) voltage compensation in combined fault condition. (a) Source current (b) Injected current (c) Load current. (a) Source voltage (b) Injected voltage (c) Load voltage.

which are appropriately selected to improve the working performance, but sometimes tuning is not appropriate. In this case, the FOPID controller problem is resolved using the hybrid optimization technique (COA-OOA). The proposed approach was analyzed under various conditions like swell, flicker, sag, interruption, and THD. The proposed model provides

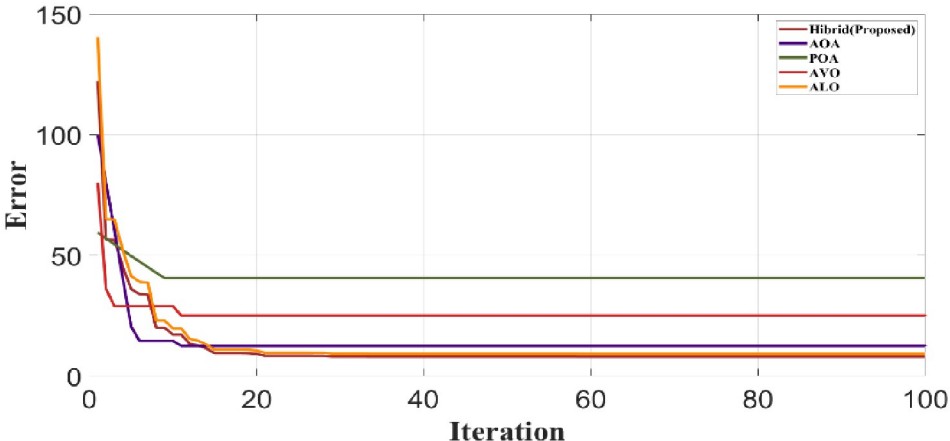

**Fig 44. Analysis of convergence comparison.**

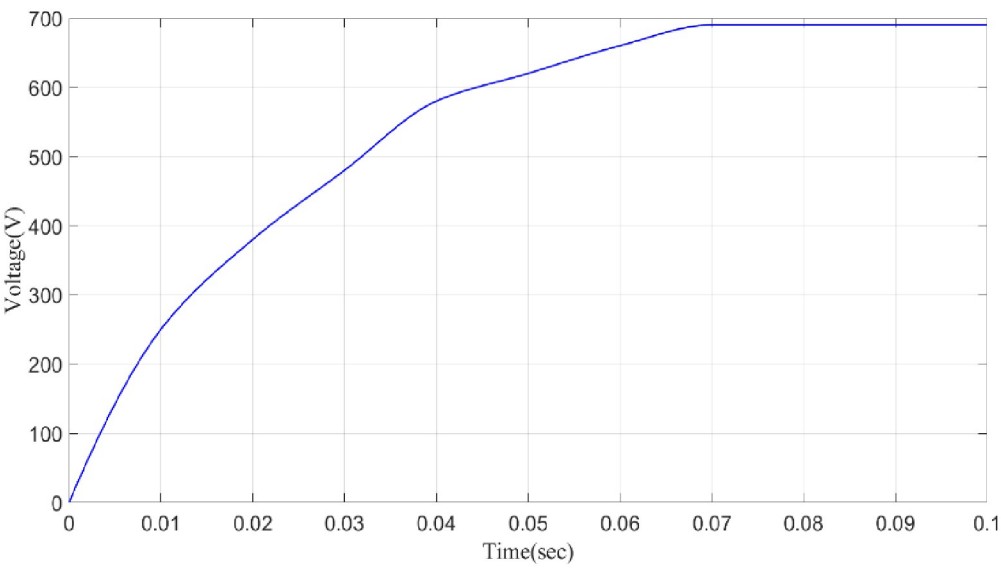

**Fig 45. Analysis of DC link voltage comparison.**

**Table 5. Analysis of power factor comparison.**

| Technique | P.F | | | | |
|---|---|---|---|---|---|
| | Case 1 | Case 2 | Case 3 | Case 4 | Case 5 |
| Without UPQC | 0.85 | 0.983 | 0.990 | 0.997 | 0.883 |
| PI-UPQC | 0.994 | 0.994 | 0.995 | 0.995 | 0.898 |
| FOPID-UPQC | 0.996 | 0.997 | 0.997 | 0.998 | 0.991 |

**Table 6. Test cases considered under constant 25°C temperature.**

| Condition/load | Case 1 | Case 2 | Case 3 | Case 4 | Case 5 |
|---|---|---|---|---|---|
| Irradiation $1000^W/_{m^2}$ | ✓ | | ✓ | | |
| Irradiation $800^W/_{m^2}$ | | ✓ | | ✓ | |
| Linear load | ✓ | ✓ | | | |
| Nonlinear load | | | ✓ | ✓ | ✓ |
| Voltage sag, swell and interruption | ✓ | ✓ | ✓ | ✓ | ✓ |
| Linear RLC load R:100 L:0.01H C:100F | ✓ | ✓ | | | |
| Nonlinear RL load R:200Ω L:0.01H | | | ✓ | ✓ | |
| Load variation in RL R:150Ω L:0.02H | | | | | ✓ |

**Table 7. THD comparison of proposed and existing methods.**

| Controller | Case 1 | Case 2 | Case 3 | Case 4 | Case 5 |
|---|---|---|---|---|---|
| Without UPQC | 19.54% | 14.30% | 11.63% | 18.63% | 25.12% |
| PI | 3.97% | 5.10% | 3.95% | 4.95% | 5.15% |
| SMC | 4.04% | 3.30% | 4.02% | 4.06% | 4.10% |
| FLC | 3.65% | 2.95% | 2.95% | 4.00% | 3.95% |
| Proposed | 0.32% | 0.11% | 12.15% | 10.11% | 10.11% |

current and voltage THD is 0.32% and 0.23% at the sag period, 0.02% and 0.71% at the swell period, and 0.98% and 0.69% at the interruption period of linear load condition. Likewise, 12.15% and 0.82% at the sag period, 10.18% and 0.48% at the swell period, and 10.07% and 1.01% at the interruption period of non-linear load condition. The mitigation performance and THD are compared to some other existing hybrid models to validate the effectiveness of the proposed model. In future work deep learning based classifier is used to tune the FOPID parameter in an effective way.

## Author Contributions

**Conceptualization:** T. Anuradha Devi, G. Srinivasa Rao, Vojtech Blazek.

**Data curation:** Habib Kraiem, Vojtech Blazek.

**Formal analysis:** Habib Kraiem.

**Funding acquisition:** T. Anuradha Devi, G. Srinivasa Rao, Flah Aymen, Habib Kraiem, Vojtech Blazek.

**Investigation:** T. Anuradha Devi, G. Srinivasa Rao, B. Srikanth Goud, Flah Aymen, Habib Kraiem, Vojtech Blazek.

**Methodology:** T. Anuradha Devi, G. Srinivasa Rao, B. Srikanth Goud, Ch. Rami Reddy.

**Project administration:** T. Anuradha Devi, B. Srikanth Goud, Ch. Rami Reddy, Vojtech Blazek.

**Resources:** T. Anuradha Devi, T. Anil Kumar, B. Srikanth Goud, Ch. Rami Reddy, Vojtech Blazek.

**Software:** T. Anuradha Devi, T. Anil Kumar, B. Srikanth Goud, Ch. Rami Reddy, Mbadjoun Wapet Daniel Eutyche, Claude Ziad El-Bayedh, Vojtech Blazek.

**Supervision:** T. Anuradha Devi, T. Anil Kumar, Mbadjoun Wapet Daniel Eutyche, Claude Ziad El-Bayedh, Vojtech Blazek.

**Validation:** T. Anil Kumar, Mbadjoun Wapet Daniel Eutyche, Claude Ziad El-Bayedh, Vojtech Blazek.

**Visualization:** T. Anil Kumar, Mbadjoun Wapet Daniel Eutyche, Claude Ziad El-Bayedh, Habib Kraiem, Vojtech Blazek.

**Writing – original draft:** G. Srinivasa Rao, T. Anil Kumar, Mbadjoun Wapet Daniel Eutyche, Claude Ziad El-Bayedh, Habib Kraiem.

**Writing – review & editing:** G. Srinivasa Rao, T. Anil Kumar, Ch. Rami Reddy, Flah Aymen, Habib Kraiem, Vojtech Blazek.

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
