## [Decision Letter · Decision Letter 0]

16 Jan 2024

PONE-D-23-43516Hybrid optimal-FOPID Based UPQC for Lessening Harmonics and Compensate Load Power in Grid Associated Renewable Energy SourcesPLOS ONE

Dear Dr. Ch Rami Reddy,

Thank you for submitting your manuscript to PLOS ONE. After careful consideration, we feel that it has merit but does not fully meet PLOS ONE’s publication criteria as it currently stands. Therefore, we invite you to submit a revised version of the manuscript that addresses the points raised during the review process.

**ACADEMIC EDITOR: **The reviewers recommend reconsideration the manuscript with revision and modification. I invite the authors to resubmit the manuscript after addressing the comments raised by the reviewers.

We look forward to receiving your revised manuscript.

Kind regards,

Dhanamjayulu C, Ph.D & Post.Doc

Academic Editor

PLOS ONE

Journal Requirements:

Additional Editor Comments (if provided):

The reviewers recommend reconsideration the manuscript with revision and modification. I invite the authors to resubmit the manuscript after addressing the comments raised by the reviewers.

Reviewers' comments:

Reviewer's Responses to Questions

**Comments to the Author**

1. Is the manuscript technically sound, and do the data support the conclusions?

Reviewer #1: Yes

Reviewer #2: Yes

Reviewer #3: Yes

Reviewer #4: Yes

2. Has the statistical analysis been performed appropriately and rigorously? 

Reviewer #1: Yes

Reviewer #2: No

Reviewer #3: Yes

Reviewer #4: Yes

3. Have the authors made all data underlying the findings in their manuscript fully available?

Reviewer #1: Yes

Reviewer #2: Yes

Reviewer #3: Yes

Reviewer #4: Yes

4. Is the manuscript presented in an intelligible fashion and written in standard English?

Reviewer #1: Yes

Reviewer #2: Yes

Reviewer #3: Yes

Reviewer #4: Yes

5. Review Comments to the Author

Reviewer #1: 1. The authors contribution of the paper must be present in bullet points and must be emphasized after introduction section

2.Included literature survey but represent it in a tabular form which looks interesting for the readers clearly

3.Provide the proposed controller parameters in a tabular form

4. The paper covers an interesting field of research. Explain what is the novelty according to already published works

5. Provide the test system simulated parameters in a table

6. What is the Role of FOPID controller how is it advanced than the traditional controller

7.Add a list of abbreviations at the end of the paper

8. Cite some latest papers in the research area of the work carried out

Reviewer #2: The authors present very interesting topic of the research about FOPID based UPQC for Power Quality improvement however few comments are suggest for the overall improvement of the work carried out

1. Abstract must be very concise and limited to the novelty. Mention the proposed optimization technique

2. Check for Fig.1 some of the parameters are missing in the block diagram

3. Flow chart of the proposed system can be improved with a better Quality

4. Include the structure of FOPID controller after the text and also provide the gains of the proposed controllers in a table

5. cite all the mathematical equations in the text

6. Proof reading is required

7. The following works will help the authors:10.1109/ACCESS.2023.3317980, https://doi.org/10.3390/su151813716, https://doi.org/10.3390/s23167146, https://doi.org/10.3390/axioms12050420, https://doi.org/10.3390/su15065209,

https://doi.org/10.1155/2022/4242996

8. Title can little bit modified as word lessening can be avoided and reframed with the other.

Reviewer #3: Very good work and flow of the paper was excellent.

A small suggestion, include comparision tables after the graph of every cases will helpful which method is best.

Some of the figures are not clear , e.g.fig.46, fig,47

Some of the figures and references not indexed inside the text.

Reviewer #4: the idea is good written but it needs further improvements like:

1. The contribution is not illustrated in an introduction. Make it in points.

2. Implementation of optimization techniques utilized is not validated. Explain in detail how to implement your model with optimization.

3. Make a deeply comparative analysis between different controllers and different optimization techniques in terms of objective function, control parameters, and computation times.

4. Articles may be used in literature like:

10.1109/ACCESS.2023.3317980

10.3390/axioms12050420

10.3390/su15065209

10.3390/su15043710

6. PLOS authors have the option to publish the peer review history of their article (what does this mean?). If published, this will include your full peer review and any attached files.

Reviewer #1: No

Reviewer #2: No

Reviewer #3: **Yes: **Dr.K.Srinivasan

Reviewer #4: No

---

## [Author Response · Author response to Decision Letter 0]

17 Feb 2024

Reviewer 1

 The authors contribution of the paper must be present in bullet points and must be emphasized after introduction section

Response: We thank the reviewer for the comment we have presented the key points of the proposed system after the introduction section in the updated manuscript. We are adding the same here for your reference.

 A novel hybrid optimization-based FOPID controller was developed to improve PQ in integrated microgrid systems in order to mitigate these problems. That is both the Coati Optimization Algorithm (COA) and Osprey Optimization Algorithm (OOA) are combined to make a hybrid optimization CO-OA algorithm. Hybrid optimization reduces the time of the decision-making process and allows to focus their time on the analyses. 

 Designed to increase PQ in a hybrid RES-grid connected non-linear distribution system. A hybrid RES based Grid with a load model was designed as per specific ranges, in between load and hybrid RES an UPQC is linked for PQ management. 

 The pulse generation of UPQC compensator's switches is done through the use of the FOPID controller, which compares the actual value and reference value to generate the pulse signal of the switches. FOPID controller's parameters are tuned by using an innovative hybrid optimization algorithm. 

 A low error function is used to choose the best optimal values. The proposed mitigation performance is analysed under sag, harmonics, interruption, and swell conditions. Moreover, the outcomes are compared to some other existing models to confirm the efficacy of the proposed model

2.Included literature survey but represent it in a tabular form which looks interesting for the readers clearly

Response: We thank the reviewer for the valuable suggestion made. We have updated the literature survey and also represented in the tabular form. 

3.Provide the proposed controller parameters in a tabular form

Response: We have updated the controller parameters as per the suggestion made. For a quiz glance we have added the table for your reference. 

Table. Controller parameters I mean gains

Parameter Ranges

µ 0.0455

KD 0.0427

KI 2.9161

KP 0.0284

λ 0.0313

4. The paper covers an interesting field of research. Explain what is the novelty according to already published works

Response: In the proposed methodology, the HRES system is utilized to provide essential power to meet load demand in load side. The first contribution provides the information about system design so only, I mentioned in contribution part. And, the UPQC device is used to compensate power quality issues in the system. Similarly, the second contribution is the objective of proposed method which used for mitigating power quality issues. 

The integration of RES in UPQC dc link capacitor is increase in the system which solved by applying control techniques of series and shunt active power filter. The control techniques were adjusting the series and shunt units to manage the dc link capacitor of UPQC. The unstable condition of UPQC may produce the stability issues because of this scenario condition, the UPQC dc link capacitor also maintain through the series and shunt active power filter control strategy.

The novelty of the work is designing a UPQC-PQ and PQ theory controller to enhance the PQ issues in the system. To enhance the PQ theory controller, FOPID controller with hybrid optimization technique is initialized which provides the best control parameters. Based on controller parameters, the stable operation and PQ issues are mitigated in the system

5. Provide the test system simulated parameters in a table

Response: We have updated test system parameters as per the suggestion made. For a quiz glance we have added the table for your reference.

Parameter Specification

DC link voltage 700-800V

Linear load 4kw

Non-linear load 6kw

Battery 4 kW

PV 214 W/panel

[I = 5.9A, V = 60V]

Irradiance 1000w/m^2

Temperature 〖25〗^0

Grid 4.375 kW

[I = 35A, V = 125V]

6. What is the Role of FOPID controller how is it advanced than the traditional controller

Response: 

Following steps are required during the design of the controller:

1. KP is regulated for minimizing steady-state error and rise time

2. Kd is regulated for minimizing the settling time and overshoot

3. Ki is regulated for eliminating the steady-state error.

4. μ,λ are fractional order parameters

7. Add a list of abbreviations at the end of the paper

Response: We thank the reviewer suggestion. We have abbreviated the terms at the first usage in the text instead as the keeping separately. 

8. Cite some latest papers in the research area of the work carried out

Response: As per the comment made we have updated latest research papers in the updated manuscripts

 Ganesan, S., David, P. W., Balachandran, P. K., & Colak, I. (2024). Power enhancement in PV arrays under partial shaded conditions with different array configuration. Heliyon.

 Srilakshmi, K., Rao, G. S., Swarnasri, K., Inkollu, S. R., Kondreddi, K., Balachandran, P. K., & Colak, I. (2024). Optimization of ANFIS controller for solar/battery sources fed UPQC using an hybrid algorithm. Electrical Engineering, 1-28.

Reviewer 2

The authors present very interesting topic of the research about FOPID based UPQC for Power Quality improvement however few comments are suggest for the overall improvement of the work carried out

1. Abstract must be very concise and limited to the novelty. Mention the proposed optimization technique

Response: We thank the reviewer for the valuable suggestion made. We have rewritten the abstract and also included the optimization technique used to tune the FOPID controller designed for mitigating the Power quality issues. 

2. Check for Fig.1 some of the parameters are missing in the block diagram

Response: As per the reviewer suggestion we have modified the block diagram of the proposed system in the updated manuscript 

3. Flow chart of the proposed system can be improved with a better Quality

Response: We have replaced the flow chart with a better Quality 

4. Include the structure of FOPID controller after the text and also provide the gains of the proposed controllers in a table

Response: As per the suggestion made we have included the FOPID controller diagram as well as controller gains in Table.

5. cite all the mathematical equations in the text

Response: We thank the reviewer for the suggestion. We have cited all the mathematical equations in the updated manuscript 

6. Proof reading is required

We thank the reviewer for the suggestion. We have done the proof read and also verified grammar with an English expert

7. The following works will help the authors:10.1109/ACCESS.2023.3317980, https://doi.org/10.3390/su151813716, https://doi.org/10.3390/s23167146, https://doi.org/10.3390/axioms12050420, https://doi.org/10.3390/su15065209,

https://doi.org/10.1155/2022/4242996

Response: we have included the following manuscript which helped authors to update the manuscript with latest papers

8. Title can little bit modified as word lessening can be avoided and reframed with the other.

Response: Title has been modified in the updated manuscript

Reviewer 3

Very good work and flow of the paper was excellent.

A small suggestion, include comparison tables after the graph of every cases will helpful which method is best.

Response: We have included the comparative values in the tabular form in terms of THD and Power factors. The other results values are presented in the text because tabular forms increases in the manuscript.

Some of the figures are not clear , e.g.fig.46, fig,47 Some of the figures and references not indexed inside the text.

Response: We thank the reviewer for the suggestion made. We have updated the manuscript with good quality figures and also cited in the updated manuscript

Reviewer 4

The idea is good written but it needs further improvements like:

1. The contribution is not illustrated in an introduction. Make it in points.

Response: We thank the reviewer for the suggestion made. We have now included the contribution of the work in bullet point after the introduction section as below

A novel hybrid optimization-based FOPID controller was developed to improve PQ in integrated microgrid systems in order to mitigate these problems. That is both the Coati Optimization Algorithm (COA) and Osprey Optimization Algorithm (OOA) are combined to make a hybrid optimization CO-OA algorithm. Hybrid optimization reduces the time of the decision-making process and allows to focus their time on the analyses. 

Designed to increase PQ in a hybrid RES-grid connected non-linear distribution system. A hybrid RES based Grid with a load model was designed as per specific ranges, in between load and hybrid RES an UPQC is linked for PQ management. 

The pulse generation of UPQC compensator's switches is done through the use of the FOPID controller, which compares the actual value and reference value to generate the pulse signal of the switches. FOPID controller's parameters are tuned by using an innovative hybrid optimization algorithm. 

A low error function is used to choose the best optimal values. The proposed mitigation performance is analyzed under sag, harmonics, interruption, and swell conditions. Moreover, the outcomes are compared to some other existing models to confirm the efficacy of the proposed model.

2. Implementation of optimization techniques utilized is not validated. Explain in detail how to implement your model with optimization.

Response: We thank the reviewer for the following comment. We have validated the Controller gains/parameter in the Table. 

Hybrid optimization parameters

Parameter Ranges 

Iteration 100

Dimension 5

T 0.01

Search agent 30

Table. 1 Controller parameters I mean gains

Parameter Ranges

µ 0.0455

KD 0.0427

KI 2.9161

KP 0.0284

λ 0.0313

3. Make a deeply comparative analysis between different controllers and different optimization techniques in terms of objective function, control parameters, and computation times.

Response: We thank the reviewer for the suggestion made. The following figures clearly shows the comparison of convergence in proposed and existing algorithms like the Aquila Optimization Algorithm (AOA), Pelican Optimization Algorithm (POA), African Vulture Optimization (AVO) and Ant Lion Optimization (ALO) Algorithm. If the relevant sequence converges for specified initial approximations, an iterative process is said to be convergent. 

Figure 46: Analysis of convergence comparison

In the proposed hybrid algorithm, the 18th iteration to overcome the error is 130. In existing methods, AOA attains the 10th error state in the 7th iteration, POA attains the 40th error state in the 5th iteration, AVO attains the 35th error state in the 10th iteration and ALO attains the 9th error in the 15th iteration. Compared to existing methods, the proposed method effectively performs.

4. Articles may be used in literature like:

10.1109/ACCESS.2023.3317980

10.3390/axioms12050420

10.3390/su15065209

10.3390/su15043710

Response: we thank the reviewer for sharing the recent papers in the same area of the manuscript which added more inputs in revising the manuscript

---

## [Editor Report · Decision Letter 1]

23 Feb 2024

Hybrid Optimal-FOPID Based UPQC for Reducing Harmonics and Compensate Load Power in Renewable Energy Sources Grid connected System

PONE-D-23-43516R1

Dear Dr. Ch Rami Reddy,

We’re pleased to inform you that your manuscript has been judged scientifically suitable for publication and will be formally accepted for publication once it meets all outstanding technical requirements.

Kind regards,

Dhanamjayulu C, Ph.D & Post.Doc

Academic Editor

PLOS ONE

Additional Editor Comments (optional):

The authors have revised the properly for reviewers concerns
---

## [Editor Report · Acceptance letter]

12 Mar 2024

PONE-D-23-43516R1 

PLOS ONE

Dear Dr. Rami Reddy, 

I'm pleased to inform you that your manuscript has been deemed suitable for publication in PLOS ONE. Congratulations! Your manuscript is now being handed over to our production team.

Kind regards, 

on behalf of

Dr. Dhanamjayulu C 

Academic Editor

PLOS ONE